# IntSGD: Adaptive Floatless Compression of Stochastic Gradients

**Konstantin Mishchenko**
CNRS, École Normale Supérieure, Inria
konsta.mish@gmail.com

**Bokun Wang**
KAUST[*]
bokunw.wang@gmail.com

**Dmitry Kovalev**
KAUST
dakovalev1@gmail.com

**Peter Richtárik**
KAUST
peter.richtarik@kaust.edu.sa

## Abstract

We propose a family of adaptive integer compression operators for distributed Stochastic Gradient Descent (SGD) that do not communicate a single float. This is achieved by multiplying floating-point vectors with a number known to every device and then rounding to integers. In contrast to the prior work on integer compression for SwitchML by Sapio et al. (2021), our IntSGD method is provably convergent and computationally cheaper as it estimates the scaling of vectors adaptively. Our theory shows that the iteration complexity of IntSGD matches that of SGD up to constant factors for both convex and non-convex, smooth and non-smooth functions, with and without overparameterization. Moreover, our algorithm can also be tailored for the popular all-reduce primitive and shows promising empirical performance.

## 1 Introduction

Many recent breakthroughs in machine learning were made possible due to the introduction of large, sophisticated and high capacity supervised models whose training requires days or even weeks of computation (Hinton et al., 2015; He et al., 2016; Huang et al., 2017; Devlin et al., 2018). However, it would not be possible to train them without corresponding advances in parallel and distributed algorithms capable of taking advantage of modern hardware. Very large models are typically trained on vast collections of training data stored in a distributed fashion across a number of compute nodes that need to communicate throughout the training process. In this scenario, reliance on efficient communication protocols is of utmost importance.

**Communication in distributed systems.** The training process of large models relies on fast synchronization of gradients computed in a parallel fashion. Formally, to train a model, we want to solve the problem of parallel/distributed minimization of the average of $n$ functions:

$$\min_{x \in \mathbb{R}^d} \left[ f(x) \stackrel{\text{def}}{=} \frac{1}{n} \sum_{i=1}^{n} f_i(x) \right], \qquad f_i(x) \stackrel{\text{def}}{=} \mathbb{E}_{\xi}[f_i(x; \xi)], \tag{1}$$

where we will compute the gradients of stochastic realizations $f_i(x; \xi)$. The two dominating protocols for gradient synchronization are *all-reduce* and *all-gather* aggregation, which may use either Parameter Server or all-to-all communication under the hood. The core difference between them lies in that all-gather communicates all vectors, whereas all-reduce only outputs their average. As shown in previous works, current distributed deep learning algorithms predominantly use all-reduce as it scales much better than all-gather (Vogels et al., 2019; Agarwal et al., 2021).

A popular way to reduce the communication cost of both all-reduce and all-gather primitives is to use lossy compression of gradients (Ramesh et al., 2021). To study the benefit of lossy compression, large swaths of recent literature on distributed training attribute the cost of sending a single vector

---

[*]Work done when the author was a research intern at KAUST.

from a worker to the server to the number of bits needed to represent it. Based on this abstraction, various elaborate vector compression techniques (see Table 1 in Beznosikov et al. 2020; Xu et al. 2020; Safaryan et al. 2020) and algorithms have been designed for higher and higher compression ratios. However, in real systems, the efficiency of sending a vector is not fully characterized by the number of bits alone, because:

- First, many compressors with high compression ratio (e.g., natural compression (Horváth et al., 2019), quantization (Alistarh et al., 2017), top-$k$ sparsification, sign (Bernstein et al., 2018)) are not compatible with the efficient all-reduce primitive and require all-gather implementation.

- Secondly, some compressors rely on expensive operations such as low-rank decomposition (Wang et al., 2018; Vogels et al., 2019) or bit-level operations (Horváth et al., 2019), whose computation overhead may outweigh the benefits of reduced communication load.

- Thirdly, algorithms with biased compressors such as Top-$k$ SGD, SignSGD, PowerSGD (Vogels et al., 2019), require the error-feedback (EF-SGD) mechanism (Stich et al., 2018; Karimireddy et al., 2019) to ensure the convergence. Alas, error feedback needs extra sequences that may not fit the low memory budget of GPUs. Moreover, to the best of our knowledge, no convergence guarantee has been established for EF-SGD on the non-smooth objectives with multiple workers.

**SwitchML.** Another approach to combating long communication times is to improve the hardware itself. The recently proposed SwitchML is an alternative to the NVIDIA Collective Communications Library (NCCL) on real-world hardware (Sapio et al., 2021). The first key component of SwitchML is the in-network aggregation (INA) by a programmable switch. INA reduces the communication cost and latency because the execution can be paralleled and pipelined. To be specific, it splits the vector to aggregate into chunks and processes them individually by the switch pipeline. The advantages of INA over parameter server and all-reduce in terms of latency and communication cost have been theoretically and empirically justified by Sapio et al. (2021). The second key component of SwitchML is stochastic gradient descent with integer rounding and aggregation. Instead of reducing the data volume to exchange, the goal of integer rounding in SwitchML is to fit the limited computation capability of the modern programmable switch, which only supports integer additions or logic operations. To increase the rounding precision, the gradient $g_i^k$ on device $i$ multiplies a positive scaling factor $\alpha_k$ known to every worker and then rounded to an integer number $\mathcal{I}nt(\alpha_k \circ g_i^k)$. As there is no additional scaling or decompression before aggregating the communicated vectors, their sums can be computed on the fly. Then, each worker can divide the aggregated gradient by $n\alpha_k$ to update the model.

However, Sapio et al. (2021) remark that the choice of the scaling factor $\alpha_k$ requires special care. In their presentation[1], one of the authors notes: "A bad choice of scaling factor can reduce the performance." To this end, they propose a heuristic-based profiling step that is executed before the gradient aggregation and keeps the rounded integers small to fit in 32 bits. We refer to their algorithm including the profiling step as Heuristic IntSGD. Unfortunately, no convergence guarantee for that algorithm has been established. This is where our theory comes to the rescue. By rigorously and exhaustively analyzing integer rounding based on scaling, we find adaptive rules for the scaling factor $\alpha_k$ that do not require the profiling employed by Sapio et al. (2021). As we will show in the remainder of the paper, our algorithm is perfectly suited for both in-network aggregation (INA) of SwitchML and for other efficient primitives such as all-reduce.

## 1.1 CONTRIBUTIONS

We summarize the key differences of our algorithm and prior work in Table 1, and we also list our main contributions below.

● **Adaptive IntSGD.** We develop a family of computationally cheap adaptive scaling factors for provably convergent IntSGD. It is a better alternative to the Heuristic IntSGD in Sapio et al. (2021) that requires expensive operations and does not ensure convergence.

---

[1] https://youtu.be/gBPHFyBWVoM?t=606

Table 1: Conceptual comparison of our method to the related literature. If all-reduce is supported, the method does not need any decompression. If all-reduce is not supported, the expensive all-gather operation is required and decompression is slow. See also Section 5 for numerical comparisons.

| Algorithm | Supports all-reduce | Supports switch | Provably works | Fast compression | Works without error-feedback | Adaptive | Reference |
|---|---|---|---|---|---|---|---|
| IntSGD | ✓ | ✓ | ✓ | ✓ | ✓ | ✓ | **Ours** |
| Heuristic IntSGD | ✓ | ✓ | ✗ | ✓ | ✓ | ✗ | Sapio et al. (2021) |
| PowerSGD (theoretical) | ✓ | ✗ | ✓ | ✗[1] | ✗ | ✗[2] | Vogels et al. (2019) |
| PowerSGD (practical) | ✓ | ✗ | ✗ | ✓[1] | ✗ | ✗[2] | Vogels et al. (2019) |
| NatSGD | ✗ | ✓ | ✓ | ✗ | ✓ | N/A | Horváth et al. (2019) |
| QSGD | ✗ | ✗ | ✓ | ✓ | ✓ | N/A | Alistarh et al. (2017) |
| SignSGD | ✗ | ✗ | ✓ | ✓ | ✗ | N/A | Karimireddy et al. (2019) |

[1] In theory, PowerSGD requires computing low-rank decompositions. In practice, an approximation is found by power iteration, which requires just a few matrix-vector multiplications but it is not analyzed theoretically and might be less stable.

[2] PowerSGD requires tuning the rank of the low-rank decomposition. Vogels et al. (2019) reported that rank-1 PowerSGD consistently underperformed in their experiments, and, moreover, rank-2 was optimal for image classification while language modeling required rank-4. Ramesh et al. (2021) reported that a much larger rank was needed to avoid a gap in the training loss.

• **Rates.** We obtain the first analysis of the integer rounding and aggregation for distributed machine learning. For all of the proposed variants, we prove convergence rates of IntSGD that match those of full-precision SGD up to constant factors. Our results are tight and apply to both convex and non-convex problems. Our analysis does not require any extra assumption compared to those typically invoked for SGD. In contrast to other compression-based methods, IntSGD has the same rate as that of full-precision SGD even on non-smooth problems.

• **IntDIANA.** We observe empirically that IntSGD struggles when the devices have heterogeneous (non-identical) data—an issue it shares with vanilla SGD—and propose an alternative method, IntDI-ANA, that can provably alleviate this issue. We also show that our tools are useful for extending the methodology beyond SGD methods, for example, to *variance reduced* methods (Johnson & Zhang, 2013; Allen-Zhu & Hazan, 2016; Kovalev et al., 2020; Gower et al., 2020) with integer rounding. Please refer to Appendix A.2 for theoretical results and Appendix C.5 for the empirical verification.

## 2 ADAPTIVE INTEGER ROUNDING AND IntSGD

By *randomized integer rounding* we mean the mapping $\mathcal{I}nt : \mathbb{R} \to \mathbb{Z}$ defined by

$$\mathcal{I}nt(t) \overset{\text{def}}{=} \begin{cases} [t] + 1, & \text{with probability } p_t \overset{\text{def}}{=} t - [t], \\ [t], & \text{with probability } 1 - p_t, \end{cases}$$

where $[t]$ denotes the floor of $t \in \mathbb{R}$, i.e., $[t] = k \in \mathbb{Z}$, where $k$ is such that $k \le t < k + 1$. Note that

$$\mathbb{E}\left[\mathcal{I}nt(t)\right] = (t - [t])([t] + 1) + ([t] + 1 - t)[t] = t.$$

We extend this mapping to vectors $x \in \mathbb{R}^d$ by applying in element-wise: $\mathcal{I}nt(x)_i \overset{\text{def}}{=} \mathcal{I}nt(x_i)$.

### 2.1 ADAPTIVE INTEGER ROUNDING

Given a *scaling vector* $\alpha \in \mathbb{R}^d$ with nonzero entries, we further define the *adaptive integer rounding* operator $Q : \mathbb{R}^d \to \mathbb{R}^d$ by

$$Q(x) \overset{\text{def}}{=} \tfrac{1}{\alpha} \circ \mathcal{I}nt(\alpha \circ x), \tag{2}$$

where $a \circ b \overset{\text{def}}{=} (a_1 b_1, \ldots, a_d b_d) \in \mathbb{R}^d$ denotes the Hadamard product of two vectors $a = (a_1, \ldots, a_d) \in \mathbb{R}^d$ and $b = (b_1, \ldots, b_d) \in \mathbb{R}^d$.

---

**Algorithm 1** IntSGD. Default setting for the tested problems: $\beta = 0.9, \varepsilon = 10^{-8}$.

---

1: **Params:** Stepsizes $\eta_k$, scaling vectors $\alpha_k \in \mathbb{R}^d$
2: **Init:** $x^0 \in \mathbb{R}^d$, $x^1 = x^0 - \eta_0 \frac{1}{n} \sum_{i=1}^n g_i^0$
3: **for** $k = 1, 2, \ldots$ **do**
4:    **for** each device $i = 1, 2, \ldots, n$ **do**
5:        Compute stochastic gradient $g_i^k$ ($\mathbb{E}\left[g_i^k \mid x^k\right] \in \partial f_i(x^k)$)
6:        Maintain the moving average: $r_k = \beta r_{k-1} + (1 - \beta)\|x^k - x^{k-1}\|^2$
7:        Compute the adaptive scaling factor: $\alpha_k = \frac{\sqrt{d}}{\sqrt{2nr_k/\eta_k^2 + \varepsilon^2}}$
8:        Scale and round the local gradient $Q(g_i^k) = \mathcal{I}nt(\alpha_k \circ g_i^k)$
9:    **end for**
10:   Aggregate $Q(g_i^k)$ by either all-reduce or in-network aggregation (INA)
11:   **for** each device $i = 1, 2, \ldots, n$ **do**
12:       Compute the (sub)gradient estimator: $\tilde{g}^k = \frac{1}{n\alpha_k} \sum_{i=1}^n Q(g_i^k)$
13:       Update the model parameter $x^{k+1} = x^k - \eta_k \tilde{g}^k$
14:   **end for**
15: **end for**

---

As we show below, the adaptive integer rounding operator (2) has several properties which will be useful in our analysis. In particular, the operator is unbiased, and its variance can be controlled by choice of a possibly random scaling vector $\alpha \in \mathbb{R}_{++}^d$.

**Lemma 1.** For any $x \in \mathbb{R}^d$ and $\alpha \in \mathbb{R}_{++}^d$, we have

$$\frac{1}{\alpha} \circ \mathbb{E}\left[\mathcal{I}nt(\alpha \circ x)\right] = x, \tag{3}$$

$$\mathbb{E}\left[\left\|\frac{1}{\alpha} \circ \mathcal{I}nt(\alpha \circ x) - x\right\|^2\right] \leq \sum_{j=1}^d \frac{1}{4\alpha_j^2}, \tag{4}$$

The expectations above are taken with respect to the randomness inherent in the rounding operator.

## 2.2 NEW ALGORITHM: INTSGD

We are ready to present our algorithm, IntSGD. At iteration $k$, each device $i$ computes a stochastic (sub)gradient vector $g_i^k$, i.e., a vector satisfying

$$\mathbb{E}\left[g_i^k \mid x^k\right] \in \partial f_i(x^k). \tag{5}$$

Prior to communication, each worker $i$ rescales its stochastic (sub)gradients $g_i^k$ using the same vector $\alpha_k \in \mathbb{R}_{++}^d$, and applies the randomized rounding operator $\mathcal{I}nt$. The resulting vectors $\mathcal{I}nt(\alpha_k \circ g_i^k)$ are aggregated to obtain $\sum_{i=1}^n \mathcal{I}nt(\alpha_k \circ g_i^k)$, which is also an integer. Each device subsequently performs division by $n$ and inverse scaling to decode the message, obtaining the vector

$$\tilde{g}^k \stackrel{\text{def}}{=} \frac{1}{n\alpha_k} \circ \sum_{i=1}^n \mathcal{I}nt(\alpha_k \circ g_i^k) = \frac{1}{n} \sum_{i=1}^n \frac{1}{\alpha_k} \circ \mathcal{I}nt(\alpha_k \circ g_i^k) \stackrel{(2)}{=} \frac{1}{n} \sum_{i=1}^n Q(g_i^k).$$

Here $\alpha_k$ is a random adaptive scaling factor calculated based on the historical information. We left the design of $\alpha_k$ to Section 4. By combining (5) and (3), we observe that $g^k$ is a stochastic (sub)gradient of $f$ at $x^k$. Finally, all devices perform in parallel an SGD-type step of the form $x^{k+1} = x^k - \eta_k \tilde{g}^k$ and the process is repeated. Our IntSGD method is formally stated as Algorithm 1 with the suggested rule of $\alpha$.

**Relation to QSGD** (Alistarh et al., 2017). QSGD bears some similarity to the IntSGD: Both of them scale $g_i^k$ by a factor before the quantization (the scaling factor in QSGD is $1/\|g_i^k\|$ for normalization). However, some key difference makes the communication efficiency of IntSGD much better than that of QSGD. It is worth noting that the normalization factors $1/\|g_i^k\|$ in QSGD are different for various workers. Then, the quantized values of various workers need to be gathered and decompressed before aggregation. On the contrary, the scaling factor $\alpha_k$ in our IntSGD is the same for all workers such that the sum of integers can be computed on the fly. Thus, IntSGD supports the efficient all-reduce primitive while QSGD does not. As seen in the experimental results in Section 5 , this

makes a big difference in empirical performance. Moreover, the proof technique for IntSGD is also intrinsically different from that of QSGD. Please see the next section for the details.

## 3  ANALYSIS OF INTSGD[2]

To establish convergence of IntSGD, we introduce the following assumption on the scaling vector $\alpha_k = (\alpha_{k,1}, \ldots, \alpha_{k,d})^\top \in \mathbb{R}^d_{++}$.

**Assumption 1.** There exists $\beta \in [0, 1)$ and a sufficiently small $\varepsilon > 0$ such that $\sum_{j=1}^d \mathbb{E}\left[\frac{\eta_k^2}{\alpha_{k,j}^2}\right]$ is

bounded above by $\eta_k^2 \varepsilon^2 + 2n(1-\beta) \sum_{t=0}^{k-1} \beta^t \mathbb{E}\left[\|x^{k-t} - x^{k-t-1}\|^2\right]$.

While this assumption may look exotic, it captures precisely what we need to establish the convergence of IntSGD, and it holds for several practical choices of $\alpha_k$, including the one shown in Section 4 and more choices in Appendix A.1.

**Challenges in IntSGD analysis.** Although the $\mathcal{I}nt$ operation is unbiased and has finite variance as shown in Lemma 1, we highlight that it is non-trivial to obtain the convergence analysis of IntSGD and the analysis is different from that of QSGD and similar methods. Indeed, QSGD, Rank-$k$, and NatSGD all use unbiased operators $\mathcal{Q}$ with variance satisfying $\mathbb{E}[\|\mathcal{Q}(g) - g\|^2] \leq \omega\|g\|^2$ for some $\omega > 0$. For them, the convergence theory is simply a plug-in of the analysis of Compressed SGD (Khirirat et al., 2018). However, the integer rounding operator $\mathcal{I}nt$ does not satisfy this property, and the variance of the integer compressor will not decrease to zero when $\|g\|^2 \to 0$. Moreover, to estimate $\alpha_k$ adaptively, we use the values of past iterates, which makes its value itself random, so the analysis of Compressed SGD cannot apply. As shown in our proofs, an extra trick is required: we reserve an additional term $\sum_{t=0}^k \mathbb{E}[\|x^{t+1} - x^t\|^2]$ to control the variance of the rounding. Furthermore, the analysis of moving-average estimation is particularly challenging since the value of $\alpha_k$ is affected by all past model updates, starting from the very first iteration.

### 3.1  NON-SMOOTH ANALYSIS: GENERIC RESULT

Let us now show that IntSGD works well even on non-smooth functions.

**Assumption 2.** Stochastic (sub)gradients $g_1^k, \ldots, g_n^k$ sampled at iteration $k$ satisfy the inequalities

$$\left\|\frac{1}{n}\sum_{i=1}^n \mathbb{E}_k[g_i^k]\right\|^2 \leq G^2, \qquad \frac{1}{n}\sum_{i=1}^n \mathbb{E}_k\left[\|g_i^k - \mathbb{E}_k[g_i^k]\|^2\right] \leq \sigma^2, \tag{6}$$

where the former inequality corresponds to $G$-Lipschitzness of $f$ and the latter to bounded variance of stochastic (sub)gradients.

**Theorem 1.** *Let functions $f_1, \ldots, f_n$ be convex and Assumptions 1 and 2 be satisfied. Then*

$$\mathbb{E}\left[f(\hat{x}^k) - f(x^*)\right] \leq \frac{\|x^0 - x^*\|^2 + 2\left(G^2 + \frac{\sigma^2}{n} + \frac{\varepsilon^2}{4n}\right)\sum_{t=0}^k \eta_t^2}{2\sum_{t=0}^{k-1} \eta_t},$$

*where $\hat{x}^k = \frac{1}{\sum_{t=0}^k \eta_t} \sum_{t=0}^k \eta_t x^t$ is a weighted average of iterates.*

### 3.2  SMOOTH ANALYSIS: GENERIC RESULT

We now develop a theory for smooth objectives.

**Assumption 3.** There exist constants $\mathcal{L}, \sigma_* \geq 0$ such that the stochastic gradients $g_1^k, \ldots, g_n^k$ at iteration $k$ satisfy $\mathbb{E}_k[g_i^k] = \nabla f_i(x^k)$ and

$$\mathbb{E}_k\left[\left\|\frac{1}{n}\sum_{i=1}^n g_i^k\right\|^2\right] \leq \mathcal{L}(f(x^k) - f(x^*)) + \frac{\sigma_*^2}{n}. \tag{7}$$

---

[2]In our analysis, we use the red color to highlight the extra terms coming from our integer compression, in contrast to the blue error terms, which come from SGD itself.

Assumption 3 is known as the *expected smoothness* assumption (Gower et al., 2019). In its formulation, we divide the constant term $\sigma_*^2$ by $n$, which is justified by the following proposition.

**Proposition 1** (Section 3.3 in Gower et al. 2019). *Let $f_i(x) = \mathbb{E}_\xi[f_i(x;\xi)]$, $g_i^k = \nabla f_i(x^k;\xi_i^k)$, and $f_i(\cdot;\xi)$ be convex and its gradient be $L_i$-Lipschitz for any $\xi$. Then, the second part of Assumption 3 is satisfied with $\sigma_*^2 \overset{\text{def}}{=} \frac{2}{n}\sum_{i=1}^n \mathbb{E}_\xi\left[\|\nabla f_i(x^*;\xi)\|^2\right]$ and $\mathcal{L} \overset{\text{def}}{=} 4\max_{i=1,\dots,n} L_i$.*

Gower et al. (2019) state and prove this result in a more general form, so for the reader's convenience, we provide a proof in the appendix.

**Theorem 2.** *Assume that $f$ is convex and Assumption 3 holds. If $\eta_k \leq \frac{1}{2\mathcal{L}}$ and $\hat{x}^k = \frac{1}{\sum_{t=0}^k \eta_t}\sum_{t=0}^k \eta_t x^t$ is a weighted average of iterates, then*

$$\mathbb{E}\left[f(\hat{x}^k) - f(x^*)\right] \leq \frac{\|x^0 - x^*\|^2 + 2\left(\frac{\sigma_*^2}{n} + \frac{\varepsilon^2}{4n}\right)\sum_{t=0}^k \eta_t^2}{2\sum_{t=0}^k \eta_t}.$$

**Corollary 1** (Overparameterized regime). When the model is overparameterized (i.e., the losses can be minimized to optimality simultaneously: $\sigma_* = 0$), we can set $\varepsilon = 0$ and obtain $\mathcal{O}\left(\frac{1}{k}\right)$ rate.

### 3.3 NON-CONVEX ANALYSIS: GENERIC RESULT

We now develop a theory for non-convex objectives.

**Assumption 4.** The gradient of $f$ is $L$-Lipschitz and there exists $f^{\text{inf}} \in \mathbb{R}$ such that $f^{\text{inf}} \leq f(x)$ for all $x$. Furthermore, for all $i$ and $k$ we have

$$\mathbb{E}\left[\|g_i^k - \nabla f_i(x^k)\|^2\right] \leq \sigma^2. \tag{8}$$

Our main result in the non-convex regime follows.

**Theorem 3.** *Let $f$ be $L$-smooth and let Assumption 1 hold. If $\eta_k \leq \frac{1}{2L}$ for all $k$, then*

$$\mathbb{E}\left[\|\nabla f(\hat{x}^k)\|^2\right] \leq 2\frac{f(x^0) - f^{\text{inf}} + \left(\frac{\sigma^2}{n} + \frac{\varepsilon^2}{4n}\right)\sum_{t=0}^k \eta_t^2 L}{\sum_{t=0}^k \eta_t}.$$

*where $\hat{x}^k$ is sampled from $\{x^0, \dots, x^k\}$ with probabilities proportional to $\eta_0, \dots, \eta_k$.*

### 3.4 EXPLICIT COMPLEXITY RESULTS

Having developed generic complexity results for IntSGD in the non-smooth (Section 3.1), smooth (Section 3.2) and non-convex (Section 3.3) regimes, we now derive explicit convergence rates.

**Corollary 2.** For any sequence of scaling vectors $\alpha_k$ satisfying Assumption 1, we recover the following complexities:

(i) if $f_1, \dots, f_n$ are convex, Assumption 2 is satisfied and $\eta_t = \eta = \frac{\|x^0 - x^*\|}{\sqrt{k(G^2 + \sigma^2/n)}} = \mathcal{O}\left(\frac{1}{\sqrt{k}}\right)$ for $t = 0, \dots, k$, then

$$\mathbb{E}\left[f(\hat{x}^k) - f(x^*)\right] = \mathcal{O}\left(\frac{\sigma + \varepsilon}{\sqrt{kn}} + \frac{G}{\sqrt{k}}\right); \tag{9}$$

(ii) if $f$ is convex, Assumption 3 holds and $\eta_t = \min\left\{\frac{1}{2\mathcal{L}}, \frac{\|x^0 - x^*\|\sqrt{n}}{\sqrt{k}(\sigma_* + \varepsilon)}\right\}$, then

$$\mathbb{E}\left[f(\hat{x}^k) - f(x^*)\right] = \mathcal{O}\left(\frac{\sigma_* + \varepsilon}{\sqrt{kn}} + \frac{\|x^0 - x^*\|}{k}\right);$$

(iii) if $f$ is non-convex, Assumption 4 holds and $\eta_t = \min\left\{\frac{1}{2L}, \frac{\sqrt{(f(x^0) - f^{\text{inf}})n}}{\sqrt{k}(\sigma + \varepsilon)}\right\}$, then

$$\mathbb{E}\left[\|\nabla f(\hat{x}^k)\|^2\right] = \mathcal{O}\left(\frac{\sigma + \varepsilon}{\sqrt{kn}} + \frac{f(x^0) - f^{\text{inf}}}{k}\right).$$

Based on Corollary 2, our IntSGD has linear speed-up in case (ii) and (iii).

**Comparison with error-feedback** (EF-SGD). Distributed SGD with biased compressors (like PowerSGD, SignSGD, Top-$k$ SGD) requires the error-feedback modification to converge. In the non-convex and smooth case, EF-SGD leads to the $\mathcal{O}\left(\frac{\sigma}{\sqrt{kn}} + \left(\frac{G}{k}\right)^{2/3}\right)$ rate in Koloskova et al. (2019) when assuming the second moment of stochastic gradient is bounded by $G^2$. Compared to their result, our rate is never worse and does not require the second moment of stochastic gradient to be bounded, which is often violated in practice even for quadratic objectives and simplest neural networks. The convergence guarantee of EF-SGD for the convex and non-smooth function (Assumption 2) is even weaker: Karimireddy et al. (2019) show the $\mathcal{O}\left(\frac{\sigma}{\sqrt{\delta k}}\right)$ convergence rate of EF-SGD only for the single-worker case ($n = 1$), which is $\frac{1}{\sqrt{\delta}}$-times worse than IntSGD ($\delta$ could be fairly small, e.g., $\delta = 1/d$ in Top-1 compression). To the best of our knowledge, there is no convergence guarantee of EF-SGD for the non-smooth function when there are multiple workers. In contrast, our IntSGD has the same rate as SGD under the same set of assumptions.

## 4 DESIGN OF SCALING FACTORS

### 4.1 ADAPTIVE SCALING FACTOR WITH THE MOVING AVERAGE AND SAFEGUARD

We now present an effective rule of adaptive $\alpha_k$ (presented in Algorithm 1) that satisfies Assumption 1 for the convergence rates listed in previous section. In the appendix, we provide more options that also satisfy Assumption 1 and the proof still goes through. For simplicity, we assume that the first communication is exact, which allows us to estimate $\alpha_k$ adaptively without worrying about $\alpha_0$.

**Proposition 2.** Assumption 1 holds if we choose $\beta \in [0, 1), \varepsilon \geq 0$ and $\alpha_k = \frac{\sqrt{d}}{\sqrt{2nr_k/\eta_k^2 + \varepsilon^2}}$, where $r_k = \beta r_{k-1} + (1 - \beta)\|x^k - x^{k-1}\|^2$.

**Remark 1.** Here $\beta \in [0, 1)$ is a constant factor to control the moving average update of $r_k$, which prevents the scaling factor $\alpha_k$ from changing too rapidly. $\varepsilon^2$ could be any sufficiently small number, which serves as a safeguard to avoid the potential "divide by zero" error. We study the sensitivity of our IntSGD to $\beta$ and $\varepsilon$ in Appendix C.4.

### 4.2 COMPRESSION EFFICIENCY

Let us now discuss the number of bits needed for the compressed vectors. Although the main attraction of IntSGD is that it can perform efficient in-network communication, we may also hope to gain from the smaller size of the updates.

Consider for simplicity the case where $\|x^k - x^{k-1}\| \approx \|\eta_k g_i^k\|$ with some $i$. The adaptive scheme with $\beta = 0$, $\varepsilon = 0$ gives $\alpha_k = \frac{\eta_k \sqrt{d}}{\sqrt{2n}\|x^k - x^{k-1}\|} \approx \frac{\eta_k \sqrt{d}}{\sqrt{2n}\|x^{k+1}-x^k\|} = \frac{\sqrt{d}}{\sqrt{2n}\|g_i^k\|}$, so that $\|\alpha_k g_i^k\|_\infty = \frac{\sqrt{d}}{\sqrt{2n}} \frac{\|g_i^k\|_\infty}{\|g_i^k\|} \leq \frac{\sqrt{d}}{\sqrt{2n}}$. Since we only use signed integers, we need at most $1 + \log_2 \frac{\sqrt{d}}{\sqrt{2n}}$ bits for each coordinate. For instance, for $d \sim 10^{10}$ and $n \sim 100$, the upper bound is $1 + \log_2(\sqrt{5 \cdot 10^7}) < 14$ bits. The situation becomes even better when $\|g_i^k\| \gg \|g_i^k\|_\infty$, i.e., when the stochastic gradients are dense. This property has been observed in certain empirical evaluations for deep neural networks; see for example the study in (Bernstein et al., 2018).

## 5 EXPERIMENTS

### 5.1 SETUP

We empirically compare our IntSGD algorithm with several representative and strong baselines: SGD, Heuristic IntSGD (Sapio et al., 2021), SGD, PowerSGD + Error-feedback (EF) (Vogels et al., 2019), NatSGD (Horváth et al., 2019), and QSGD (Alistarh et al., 2017). The experiments are performed on 16 NVIDIA Tesla V100 GPUs located on 8 compute nodes of a cluster (2 GPUs per node) following the PowerSGD paper. The compute nodes in the cluster utilize InfiniBand HDR-100 Director Switch at 100Gbps speed for network connection. The cluster also supports the NVIDIA Collective Communications Library (NCCL).

We consider two tasks: image classification by ResNet18 (He et al., 2016) on the CIFAR-10 dataset and language modeling by a 3-layer LSTM on the Wikitext-2 dataset. The neural network architectures and hyperparameters are from some public PyTorch implementations[3]. Our code is built on the codebase of PowerSGD[4]. We also borrow their all-reduce-based implementations of SGD and PowerSGD. It is worth noting that QSGD and NatSGD do not support all-reduce. Thus, we implement their collective communications by all-gather. The implementations for compression and decompression in QSGD and NatSGD are from the authors of NatSGD[5]. For the sake of comparison, we also implement the all-gather-based SGD. We report the results of 3 repetitions with varying seeds.

Apart from the IntSGD with randomized integer rounding (IntSGD (Random)) analyzed in our theory, we also consider the variant of IntSGD with deterministic integer rounding (IntSGD (Determ.)) which can use the PyTorch built-in function `torch.round`. For all IntSGD variants, we clip the local stochastic gradients to ensure that each aggregated value fits in either 8 bits or 32 bits.

For more details of the experimental setup, please refer to Appendix C.1.

## 5.2 IntSGD vs. Heuristic IntSGD

First, we compare our IntSGD with the most related algorithm Heuristic IntSGD (Sapio et al., 2021). For both algorithms, we consider two potential communication data types: `int8` and `int32`. Note that the rule of scaling factor in Heuristic IntSGD is $\alpha = \frac{2^{nb}-1}{n \cdot 2^{max\_exp}}$, where "nb" represents the number of bits to encode each coordinate and "max_exp" is the rounded exponent of the largest absolute value in the communicated package. Although this scaling rule is straightforward and avoids overflows, it cannot guarantee convergence, even with `int32` as the communication data type. Indeed, the Heuristic IntSGD may fail to match the testing performance of full-precision SGD according to Figure 1. On the contrary, our IntSGD can perfectly match the performance of full-precision SGD on both image classification and language modeling tasks, which is in accordance with our theory that IntSGD is provably convergent.

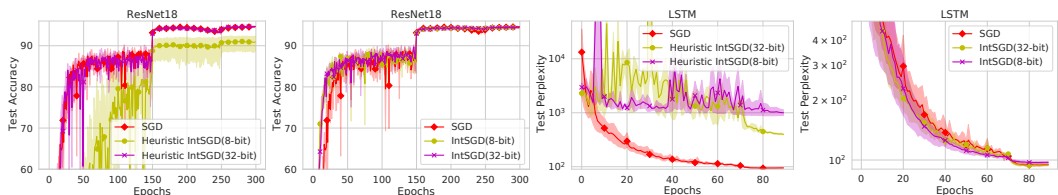

Figure 1: Comparison among IntSGD (8-bit or 32-bit), Heuristic IntSGD (8-bit or 32-bit), and full-precision SGD on the tasks of training ResNet18 and LSTM.

## 5.3 IntSGD vs. other baselines

We also compare our IntSGD algorithms to the other baselines including the all-gather-based SGD, QSGD, NatSGD and the all-reduce-based SGD, PowerSGD (EF) on the two tasks. See the test performance and time breakdown in Table 2 and Table 3.

First, we can observe that the all-gather based SGD + compressors (e.g., QSGD, NatSGD) are indeed faster than the all-gather based full-precision SGD, which shows the benefit of lossy compressions. However, they are even much slower than the all-reduce-based full-precision SGD. Unfortunately, QSGD and NatSGD) does not support the more efficient all-reduce primitive. Similar observation can be seen in previous works (Vogels et al., 2019; Agarwal et al., 2021).

All of PowerSGD (EF), IntSGD (Random), and IntSGD (Determ.) are consistently faster than the all-reduce-based full-precision SGD on both tasks. Compared to IntSGD (Determ.), IntSGD (Random) leads to slightly more computation overhead due to the randomized rounding. However, IntSGD

---

[3]ResNet18: https://github.com/kuangliu/pytorch-cifar; LSTM: https://github.com/pytorch/examples/tree/master/word_language_model
[4]https://github.com/epfml/powersgd
[5]https://github.com/sands-lab/grace

Table 2: Test accuracy and time breakdown in one iteration (on average) of training ResNet18 on the CIFAR-10 dataset with 16 workers. All numbers of time are in millisecond (ms). In each column, the best one is highlighted in black and the second-best one is highlighted in gray.

| Algorithm | Test Accuracy (%) | Computation Overhead | Communication | Total Time |
|---|---|---|---|---|
| SGD (All-gather) | $94.65 \pm 0.08$ | - | $261.29 \pm 0.98$ | $338.76 \pm 0.76$ |
| QSGD | $93.69 \pm 0.03$ | $129.25 \pm 1.58$ | $138.16 \pm 1.29$ | $320.49 \pm 2.11$ |
| NatSGD | $94.57 \pm 0.13$ | $36.01 \pm 1.30$ | $106.27 \pm 1.43$ | $197.18 \pm 0.25$ |
| SGD (All-reduce) | $94.67 \pm 0.17$ | - | $18.48 \pm 0.09$ | $74.32 \pm 0.06$ |
| PowerSGD (EF) | $94.33 \pm 0.15$ | $7.07 \pm 0.03$ | $5.03 \pm 0.07$ | $67.08 \pm 0.06$ |
| IntSGD (Determ.) | $94.43 \pm 0.12$ | $2.51 \pm 0.04$ | $6.92 \pm 0.07$ | $64.95 \pm 0.15$ |
| IntSGD (Random) | $94.55 \pm 0.13$ | $3.20 \pm 0.02$ | $6.21 \pm 0.13$ | $65.22 \pm 0.08$ |

(Determ.) fails to match the testing performance of SGD on the language modeling task. Compared to PowerSGD (EF), our IntSGD variants are better on the task of training ResNet18 but inferior on the task of training a 3-layer LSTM. Although IntSGD is not always better than PowerSGD (EF), there are several scenarios where IntSGD is preferrable as explained in in Section 1 and Section 3.4. In addition, as seen in Figure 3 of the Appendix C.3, PowerSGD (EF) converges much slower than SGD and IntSGD in the first 150 epochs of the ResNet training (which has non-smooth activations).

Table 3: Test loss and time breakdown in one iteration (on average) of training a 3-layer LSTM on the Wiki-text2 dataset with 16 workers. All numbers of time are in millisecond (ms). In each column, the best one is highlighted in black and the second-best one is highlighted in gray.

| Algorithm | Test Loss | Computation Overhead | Communication | Total Time |
|---|---|---|---|---|
| SGD (All-gather) | $4.52 \pm 0.01$ | - | $733.07 \pm 1.04$ | $796.23 \pm 1.03$ |
| QSGD | $4.63 \pm 0.01$ | $43.67 \pm 0.11$ | $307.63 \pm 1.16$ | $399.10 \pm 1.25$ |
| NatSGD | $4.52 \pm 0.01$ | $64.63 \pm 0.12$ | $309.87 \pm 1.32$ | $422.49 \pm 2.15$ |
| SGD (All-reduce) | $4.54 \pm 0.03$ | - | $22.33 \pm 0.02$ | $70.46 \pm 0.05$ |
| PowerSGD (EF) | $4.52 \pm 0.01$ | $4.22 \pm 0.01$ | $2.10 \pm 0.01$ | $54.89 \pm 0.02$ |
| IntSGD (Determ.) | $4.70 \pm 0.02$ | $3.04 \pm 0.01$ | $6.94 \pm 0.05$ | $57.93 \pm 0.03$ |
| IntSGD (Random) | $4.54 \pm 0.01$ | $4.76 \pm 0.01$ | $7.14 \pm 0.04$ | $59.99 \pm 0.01$ |

## 6 CONCLUSION

In this paper, we propose the provably convergent and computationally cheap IntSGD algorithm for efficient distributed machine learning. The core component of IntSGD is the adaptively estimated scaling factor shared by all users, which makes it compatible with the widely used communication primitive all-reduce and the recently proposed in-network aggregation (INA) (Sapio et al., 2021). The convergence rates of IntSGD match that of SGD up to constant factors on a broad spectrum of problems. Experimental results on two deep learning tasks show its promising empirical performance. A limitation of our algorithm is that its compression ratio is bounded by 4, but we hope to address this in a future work.

**Reproducibility statement.** Regarding the theoretical results: We describe the mathematical setting and algorithms in Section 1, 2, and Appendix A; Assumptions and the main theoretical results are presented in Section 3; We provide the complete proof for those results in Appendix B. Regarding the experimental results: We report the number of repetitions, the computing infrastructure used, the range of hyper-parameters considered, and the evaluation metrics in Section 5 and Appendix C.1; We attach our code in the supplementary material.

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

# Appendix

## A OTHER VARIANTS OF INTSGD

### A.1 OTHER CHOICES OF SCALING FACTOR $\alpha_k$

In Section 4.1, we provide an effective scaling factor with the moving average and the safeguard. However, there are other choices of scaling factor that also satisfy Assumption 1, and the convergence proof still goes through.

**Proposition 3** (Adaptive $\alpha_k$). If we choose

$$\alpha_k = \frac{\eta_k \sqrt{d}}{\sqrt{2n}\|x^k - x^{k-1}\|},$$

then Assumption 1 holds with $\varepsilon = 0$ and $\beta = 0$.

One can also consider applying an integer quantization with individual values of $\alpha_t$ for each coordinate or block, for instance, with an $\alpha_{t,l}$ corresponding to the $l$-th layer in a neural network. It is straightforward to see that this modification leads to the error $\sum_{l=1}^{B} d_l \frac{\eta_k^2}{\alpha_{k,l}^2}$, where $B$ is the total number of blocks and $d_l$ is the dimension of the $l$-th block.

**Proposition 4** (Adaptive block $\alpha_k$). Assume we are given a partition of all coordinates into $B \leq d$ blocks with dimensions $d_1, \ldots, d_B$, and denote by $(x^k)_l$ the $l$-th block of coordinates of $x^k$. Then Assumption 1 holds with

$$\alpha_{k,(l)} = \frac{\eta_k \sqrt{d_l}}{\sqrt{2n}\|(x^k)_l - (x^{k-1})_l\|}, \text{ for } l = 1, \ldots, B.$$

There are two extreme cases in terms of how we can choose the blocks. One extreme is to set $B = 1$, in which case we have a single scalar for the whole vector. The other extreme is to use $B = d$, which means that $\alpha_k = \frac{\eta_k}{2\sqrt{n}|x^k - x^{k-1}|}$, where the division and absolute values are computed coordinate-wise.

---

**Algorithm 2** IntSGD: adaptive block quantization

1: **Input:** $x^0 \in \mathbb{R}^d$, $\beta \in [0, 1)$, $\varepsilon \geq 0$, $x^1 = x^0 - \eta_0 \frac{1}{n} \sum_{i=1}^{n} g_i^0$ a partitioning of $\mathbb{R}^d$ into $B$ blocks of sizes $d_1, \ldots, d_B$ such that $\mathbb{R}^d = \mathbb{R}^{d_1} \times \cdots \times \mathbb{R}^{d_B}$
2: **for** $k = 1, 2, \ldots$ **do**
3:     **for** each device $i = 1, 2, \ldots, n$ **do**
4:         Compute independent stochastic gradients $g_i^k$ ($\mathbb{E}_k[g_i^k] \in \partial f_i(x^k)$)
5:         Maintain the exponetial moving average: $r_{k,l} = \beta r_{k-1,l} + (1-\beta)\|(x^k)_l - (x^{k-1})_l\|^2$ {for each block $l = 1, \ldots, B$}
6:         Compute the adaptive scaling factors: $\alpha_{k,l} = \frac{\eta_k \sqrt{d_l}}{\sqrt{2n r_{k,l} + \eta_k^2 \frac{d_l}{d} \varepsilon^2}}$
7:         Scale and round the local gradient $(Q(g_i^k))_l = \mathcal{I}nt(\alpha_{k,l}(g_i^k)_l)$
8:     **end for**
9:     Aggregate $Q(g_i^k)$ by either all-reduce or in-network aggregation (INA)
10:    **for** each device $i = 1, 2, \ldots, n$ **do**
11:        Compute the (sub)gradient estimator: $(\tilde{g}^k)_l = \frac{1}{n\alpha_{k,l}} \sum_{i=1}^{n}(Q(g_i^k))_l$
12:        $x^{k+1} = x^k - \eta_k \tilde{g}^k$
13:    **end for**
14: **end for**

---

**Compression efficiency of IntSGD with adaptive block quantization.** Our block-wise and coordinate-wise compression can further benefit from reduced dimension factors in the upper bounds, leading to the estimate of $\log_2 \frac{\sqrt{d_l}}{2\sqrt{n}}$ bits for block with dimension $d_l$. However, for smaller blocks it is less likely to happen that $\|(x^k)_l - (x^{k-1})_l\| \approx \|\eta_k(g_i^k)_l\|$, so the estimate should be taken

---

**Algorithm 3** IntDIANA

---

1: **Params:** Stepsizes $\eta_k$, scaling vectors $\alpha_k \in \mathbb{R}^d$
2: **Init:** $x^0 \in \mathbb{R}^d$, $x^1 = x^0 - \eta_0 \frac{1}{n} \sum_{i=1}^n g_i^0$, $h_i^1 = 0$, $h^1 = 0$
3: **for** $k = 1, 2, \ldots$ **do**
4:     **for** each device $i = 1, 2, \ldots, n$ **do**
5:         Compute stochastic gradient $g_i^k$ ($\mathbb{E}\left[g_i^k \mid x^k\right] \in \partial f_i(x^k)$).
6:         Compute the adaptive scaling factor: $\alpha_k = \frac{\eta_k \sqrt{d}}{\sqrt{2n}\|x^k - x^{k-1}\|}$
7:         Scale and round the local gradient $Q(g_i^k) = \mathcal{I}nt(\alpha_k \circ (g_i^k - h_i^k))$
8:         Update the local shift $h_i^{k+1} = h_i^k + Q(g_i^k)$
9:     **end for**
10:    Aggregate $Q(g_i^k)$ by either all-reduce or in-network aggregation (INA)
11:    **for** each device $i = 1, 2, \ldots, n$ **do**
12:        Compute the (sub)gradient estimator: $\tilde{g}^k = h^k + \frac{1}{n\alpha_k} \sum_{i=1}^n Q(g_i^k)$
13:        Update the model parameter $x^{k+1} = x^k - \eta_k \tilde{g}^k$
14:        Update global shift $h^{k+1} = h^k + \frac{1}{n\alpha_k} \sum_{i=1}^n Q(g_i^k)$
15:    **end for**
16: **end for**

---

with a grain of salt. We hypothesize that using $\varepsilon$ as in Proposition 2 is required to make block compression robust. Notice that if stochastic gradients have bounded coordinates, i.e., $\|g_i^k\|_\infty \le G_\infty$ for all $i, k$, then we would need at most $1 + \log_2 \frac{\sqrt{d}G_\infty}{\varepsilon}$ bits to encode the integers. Since any $\varepsilon \le \sigma + \sqrt{n}G$ does not change the rate in the non-smooth case (see Equation (9)), we get for free the upper bound of $1 + \log_2 \frac{\sqrt{d}G_\infty}{\sqrt{n}G}$ bits.

## A.2   Handling heterogeneous data

IntSGD can be equipped with the full gradient or variance-reduced gradient estimator to enjoy faster convergence than $\mathcal{O}\left(\frac{1}{\sqrt{k}}\right)$ shown in Corollary 2. For example, if we plug $\sigma_* = 0$ (no variance) and $\varepsilon \le \frac{\sqrt{n}}{\sqrt{k}}$ (sufficiently small safeguard) into item 2 of Corollary 2, the convergence rate of IntSGD is $\mathcal{O}\left(\frac{1}{k}\right)$. However, when the data are heterogeneous (i.e., minimizing $f(x) = \frac{1}{n} \sum_{i=1}^n f_i(x)$ will not make $\|\nabla f_i(x^*)\| = 0$, $\forall i \in [n]$), the transmitted integer of IntSGD with $\sigma_* = 0$, $\varepsilon = 0$ can be gigantically large, which leads to very inefficient communications or even exception value error. E.g., if we choose the adaptive $\alpha_k$ and the full gradient $g_i^k = \nabla f_i(x^k)$, the largest integer to transmit from worker $i$ to the master is $\|\alpha_k \nabla f_i(x^k)\|_\infty \approx \frac{\|\nabla f_i(x^k)\|_\infty}{\|x^k - x^{k-1}\|}$, where the denominator is 0 while the numerator is nonzero as the iterate converges to the optimum. To alleviate this issue, one needs to compress gradient *differences* as is done for example by Mishchenko et al. (2019) in their DIANA method. By marrying IntSGD with the DIANA trick, we obtain IntDIANA (Algorithm 3).

For IntDIANA with adaptive $\alpha_k$, the largest transmitted integer from worker to the master is $\|\alpha_k(g_i^k - h_i^k)\|_\infty \approx \frac{\|g_i^k - h_i^k\|_\infty}{\|x^k - x^{k-1}\|}$. We will show that both the nominator and the denominator are infinitesimal when $x^k$ converges to the optimum, such that the issue mentioned above can hopefully be solved.

Note that we can either use the full gradient $g_i^k = \nabla f_i(x^k)$ or the L-SVRG estimator Kovalev et al. (2020)

$$g_i^k = \nabla f_i(x^k; \xi_i^k) - \nabla f_i(w_i^k; \xi_i^k) + \mathbb{E}\left[\nabla f_i(w_i^k; \xi)\right]$$

on the $i$-th worker. For the variance-reduced method, we further assume that $f_i$ has a finite-sum structure, i.e.,

$$f_i(x) = \frac{1}{m} \sum_{l=1}^m f_{il}(x)$$

such that $\nabla f_i(x; \xi) = \nabla f_{il}(x)$, $\mathbb{E}\left[\nabla f_i(x; \xi)\right] = \frac{1}{m} \sum_{l'=1}^m \nabla f_{il'}(x)$ and $l$ is sampled from $[m]$ uniformly at random by $\xi$.

Our main convergence theorem describing the behavior of IntDIANA follows:

**Theorem 4.** *Assume that $f$ is $\mu$-strongly convex ($\mu \geq 0$) and $f(\cdot; \xi)$ has $L_i$-Lipschitz gradient for any $\xi$, $\mathcal{L} \stackrel{\text{def}}{=} 4 \max_i L_i$.*

1. *If $\mu > 0$, the iterates of IntDIANA with adaptive $\alpha_k = \frac{\eta \sqrt{d}}{\sqrt{n} \|x^k - x^{k-1}\|}$ satisfy*

$$\mathbb{E}\left[\Psi^k\right] \leq \theta^k \Psi^0.$$

- *For IntDIANA with the GD estimator $g_i^k = \nabla f_i(x^k)$, we have $\theta \stackrel{\text{def}}{=} \max\left\{1 - \eta\mu, \frac{3}{4}\right\} < 1$ and $\Psi^k \stackrel{\text{def}}{=} \|x^k - x^*\|^2 + \|x^k - x^{k-1}\|^2 + \frac{\eta^2 L^2}{4n^2} \sum_{i=1}^{n} \|h_i^k - \nabla f_i(x^*)\|^2$, where $\eta_k = \eta \leq \frac{1}{2(L + \frac{\mathcal{L}}{32n})}$;*

- *For IntDIANA with the L-SVRG estimator, we have $\theta \stackrel{\text{def}}{=} \max\left\{1 - \eta\mu, \frac{3}{4}, 1 - \frac{3}{8m}\right\} < 1$ and $\Psi^k \stackrel{\text{def}}{=} \|x^k - x^*\|^2 + \|x^k - x^{k-1}\|^2 + \frac{8\eta^2}{n^2} \sum_{i=1}^{n} \sum_{l=1}^{m} \|\nabla f_{il}(w_i^k) - \nabla f_{il}(x^*)\|^2 + \frac{\eta^2 L^2}{4n^2} \sum_{i=1}^{n} \|h_i^k - \nabla f_i(x^*)\|^2$, where $p = \frac{1}{m}$, and $\eta_k = \eta \leq \frac{1}{2(L + 2\mathcal{L}/n)}$.*

2. *If $\mu = 0$, the iterates of IntDIANA with adaptive $\alpha_k = \frac{\eta \sqrt{d}}{\sqrt{n} \|x^k - x^{k-1}\|}$ satisfy*

$$\mathbb{E}\left[f(\hat{x}^k) - f(x^*)\right] \leq \frac{\Psi^0}{\eta(k+1)},$$

*where $\hat{x}^k = \frac{1}{k+1} \sum_{i=0}^{k} x^i$.*

- *IntDIANA with the GD estimator requires that $\eta_k = \eta \leq \frac{1}{4(L + \frac{\mathcal{L}}{32n})}$,*
- *IntDIANA with the L-SVRG estimator requires that $\eta_k = \eta \leq \frac{1}{4(L + 2\mathcal{L}/n)}$.*

The above theorem establishes linear convergence of two versions of IntDIANA in the strongly convex regime and sublinear convergence in the convex regime.

**Compression efficiency of IntDIANA.** If $\mu > 0$, for IntDIANA with adaptive $\alpha_k$ and either GD or L-SVRG estimator, both $\|h_i^k - \nabla f_i(x^*)\|^2$ and $\|x^k - x^{k-1}\|^2$ converge to 0 linearly at the same rate, while $g_i^k \to \nabla f_i(x^*)$. Thus, the largest integer to transmit is $\|\alpha_k(g_i^k - h_i^k)\|_\infty \approx \frac{\|g_i^k - h_i^k\|_\infty}{\|x^k - x^{k-1}\|}$ is hopefully upper bounded.

## B  PROOFS

### B.1  PROOFS FOR IntSGD

In the section, we provide the complete convergence proof of IntSGD.

#### B.1.1  PROOFS FOR LEMMA 1

*Proof.* Take $y = \alpha \circ x$ and let $p_y = y - [y]$, where $[y]$ is the coordinate-wise floor, and $p_y$ is the vector of probabilities in the definition of $\mathcal{I}nt(y)$. By definition it holds

$$\mathbb{E}\left[\mathcal{I}nt(y)\right] = p_y([y] + 1) + (1 - p_y)[y] = p_y + [y] = y - [y] + [y] = y.$$

Plugging back $y = \alpha \circ x$, we obtain the first claim.

Similarly,

$$\|y - \mathcal{I}nt(y)\|_\infty = \max_{j=1,\ldots,d} \left|y_j - \mathcal{I}nt(y_j)\right| \leq \max_{z \in \mathbb{R}} \max\left(z - [z], [z] + 1 - z\right) = 1.$$

After substituting $y = \alpha \circ x$, it remains to mention

$$\left\|\frac{1}{\alpha} \circ \mathcal{I}nt(\alpha \circ x) - x\right\|_\infty \leq \|\mathcal{I}nt(\alpha \circ x) - \alpha \circ x\|_\infty \max_{j=1,\ldots,d} \frac{1}{\alpha_j}.$$

To obtain the last fact, notice that $\mathcal{I}nt(y) - [y]$ is a vector of Bernoulli random variables. Since the variance of any Bernoulli variable is bounded by $\frac{1}{4}$, we have

$$\mathbb{E}\left[\left\|\frac{1}{\alpha} \circ \mathcal{I}nt(\alpha \circ x) - x\right\|^2\right] = \sum_{j=1}^d \frac{1}{\alpha_j^2} \mathbb{E}\left[(\mathcal{I}nt(y_j) - y_j)^2\right] \leq \sum_{j=1}^d \frac{1}{4\alpha_j^2}.$$

$\square$

The staring point of our analysis is the following recursion. Let $\rho^k \overset{\text{def}}{=} \|x^k - x^*\|^2$, $\delta^k \overset{\text{def}}{=} f(x^k) - f(x^*)$ and $\zeta^k \overset{\text{def}}{=} \|\frac{1}{n}\sum_{i=1}^n g_i^k\|^2$.

**Lemma 2.** Assume that either i) functions $f_1, \ldots, f_n$ are convex, or ii) $f$ is convex and $f_1, \ldots, f_n$ are differentiable. Then

$$\mathbb{E}_k\left[\rho^{k+1}\right] \leq \rho^k - 2\eta_k\delta^k + A^k + B^k,$$

where $A^k \overset{\text{def}}{=} 2\eta_k^2\mathbb{E}_k\left[\zeta^k\right]$ and $B^k \overset{\text{def}}{=} \frac{1}{2n}\sum_{j=1}^d \frac{\eta_k^2}{\alpha_{k,j}^2} - \|x^{k+1} - x^k\|^2$ are the SGD and quantization error terms, respectively.

### B.1.2 PROOF OF LEMMA 2

*Proof.* The last term in the expression that we want to prove is needed to be later used to compensate quantization errors. For this reason, let us save one $\|x^{k+1} - x^k\|^2$ for later when expanding $\|x^{k+1} - x^*\|^2$. Consider the IntSGD step $x^{k+1} - x^k = \eta_k\frac{1}{n}\sum_{i=1}^n Q(g_i^k)$, where $Q(g_i^k) = \frac{1}{\alpha_k} \circ \mathcal{I}nt(\alpha_k \circ g_i^k)$.

$$\begin{aligned}
\|x^{k+1} - x^*\|^2 &= \|x^k - x^*\|^2 + 2\langle x^{k+1} - x^k, x^k - x^*\rangle + \|x^{k+1} - x^k\|^2 \\
&= \|x^k - x^*\|^2 + 2\langle x^{k+1} - x^k, x^k - x^*\rangle + 2\|x^{k+1} - x^k\|^2 - \|x^{k+1} - x^k\|^2 \\
&= \|x^k - x^*\|^2 - 2\frac{\eta_k}{n}\sum_{i=1}^n \langle Q(g_i^k), x^k - x^*\rangle + 2\eta_k^2\left\|\frac{1}{n}\sum_{i=1}^n Q(g_i^k)\right\|^2 - \|x^{k+1} - x^k\|^2.
\end{aligned}$$
(10)

Now let us forget about the first and last terms, which will directly go into the final bound, and work with the other terms. By our assumptions, we either have $\mathbb{E}_k[Q(g_i^k)] = \mathbb{E}_k[g_i^k] \in \partial f_i(x^k)$ or $\frac{1}{n}\sum_{i=1}^n \mathbb{E}_k[g_i^k] = \nabla f(x^k)$, so we obtain by convexity

$$\mathbb{E}_k\left[-2\frac{\eta_k}{n}\sum_{i=1}^n \langle Q(g_i^k), x^k - x^*\rangle\right] \overset{(3)}{=} -2\frac{\eta_k}{n}\sum_{i=1}^n \langle \mathbb{E}_k[g_i^k], x^k - x^*\rangle \leq -2\eta_k(f(x^k) - f(x^*)).$$

Moreover, using the tower property of expectation, we can decompose the penultimate term in (10) as follows:

$$\begin{aligned}
\mathbb{E}_k\left[\left\|\frac{1}{n}\sum_{i=1}^n Q(g_i^k)\right\|^2\right] &= \mathbb{E}_k\left[\mathbb{E}_Q\left[\left\|\frac{1}{n}\sum_{i=1}^n Q(g_i^k)\right\|^2\right]\right] \\
&= \mathbb{E}_k\left[\left\|\frac{1}{n}\sum_{i=1}^n \mathbb{E}_Q[Q(g_i^k)]\right\|^2 + \mathbb{E}_Q\left[\left\|\frac{1}{n}\sum_{i=1}^n (Q(g_i^k) - \mathbb{E}_Q[Q(g_i^k)])\right\|^2\right]\right] \\
&\overset{(3)}{=} \mathbb{E}_k\left[\left\|\frac{1}{n}\sum_{i=1}^n g_i^k\right\|^2\right] + \frac{1}{n^2}\sum_{i=1}^n \mathbb{E}_k\left[\|Q(g_i^k) - g_i^k\|^2\right],
\end{aligned}$$

where in the last step we also used independence of the quantization errors $(Q(g_1^k) - g_1^k), \ldots, (Q(g_n^k) - g_n^k)$.

Next, we are going to deal with the quantization terms:

$$\sum_{i=1}^n \mathbb{E}_k\left[\|Q(g_i^k) - g_i^k\|^2\right] = \sum_{i=1}^n \mathbb{E}_k\left[\left\|\frac{1}{\alpha_k} \circ \mathcal{I}nt(\alpha_k \circ g_i^k) - g_i^k\right\|^2\right] \overset{(4)}{\leq} \sum_{i=1}^n \sum_{j=1}^d \frac{1}{4\alpha_{k,j}^2} = \frac{n}{4}\sum_{j=1}^d \frac{1}{\alpha_{k,j}^2}.$$

Dividing both sides by $n^2$ and plugging it into (10), we obtain the desired decomposition into SGD and quantization terms. $\square$

We now show how to control the quantization error by choosing the scaling vector $\alpha_k$ in accordance with Assumption 1.

**Lemma 3.** If the assumptions of Lemma 2 hold together with Assumption 1, then

$$\mathbb{E}\left[\rho^{k+1}\right] \le \rho^0 - 2\sum_{t=0}^{k}\eta_t\mathbb{E}\left[\delta^t\right] + 2\sum_{t=0}^{k}\eta_t^2\mathbb{E}\left[\zeta^t\right] + \frac{\varepsilon^2}{2n}\sum_{t=1}^{k}\eta_t^2.$$

### B.1.3 PROOF OF LEMMA 3

*Proof.* Firstly, let us recur the bound in Lemma 2 from $k$ to 0:

$$\mathbb{E}\left[\|x^{k+1} - x^*\|^2\right] \le \|x^0 - x^*\|^2 - 2\sum_{t=0}^{k}\eta_t\mathbb{E}\left[f(x^t) - f(x^*)\right] + 2\sum_{t=0}^{k}\eta_t^2\mathbb{E}\left[\left\|\frac{1}{n}\sum_{i=1}^{n}g_i^t\right\|^2\right]$$

$$+ \frac{1}{2n}\sum_{t=1}^{k}\sum_{j=1}^{d}\mathbb{E}\left[\frac{\eta_k^2}{\alpha_{k,j}^2}\right] - \sum_{t=0}^{k}\mathbb{E}\left[\|x^{t+1} - x^t\|^2\right].$$

Note that in the bound we do not have $\alpha_{0,j}$ for any $j$ as we assume that the first communication is done without compression. Assumption 1 implies for the quantization error

$$\sum_{t=1}^{k}\sum_{j=1}^{d}\mathbb{E}\left[\frac{\eta_t^2}{\alpha_{t,j}^2}\right] \le \sum_{t=1}^{k}\left(\eta_t^2\varepsilon^2 + (1-\beta)\sum_{l=0}^{t-1}\beta^l\mathbb{E}\left[\|x^{t-l} - x^{t-l-1}\|^2\right]\right)$$

$$= \varepsilon^2\sum_{t=1}^{k}\eta_t^2 + (1-\beta)\sum_{t=1}^{k}\left(\mathbb{E}\left[\|x^t - x^{t-1}\|^2\right]\sum_{l=0}^{k-t}\beta^l\right)$$

$$\le \varepsilon^2\sum_{t=1}^{k}\eta_t^2 + (1-\beta)\sum_{t=1}^{k}\left(\mathbb{E}\left[\|x^t - x^{t-1}\|^2\right]\sum_{l=0}^{\infty}\beta^l\right)$$

$$= \varepsilon^2\sum_{t=1}^{k}\eta_t^2 + \sum_{t=1}^{k}\mathbb{E}\left[\|x^t - x^{t-1}\|^2\right]. \tag{11}$$

It is clear that the latter terms get canceled when we plug this bound back into the first recursion. $\quad\square$

### B.1.4 PROOF OF THEOREM 1

*Proof.* Most of the derivation has been already obtained in Lemma 3 and we only need to take care of the SGD terms. To do that, we decompose the gradient error into expectation and variance:

$$\mathbb{E}_k\left[\left\|\frac{1}{n}\sum_{i=1}^{n}g_i^k\right\|^2\right] = \left\|\frac{1}{n}\sum_{i=1}^{n}\mathbb{E}_k[g_i^k]\right\|^2 + \mathbb{E}_k\left[\left\|\frac{1}{n}\sum_{i=1}^{n}(g_i^k - \mathbb{E}_k[g_i^k])\right\|^2\right]$$

$$= \left\|\frac{1}{n}\sum_{i=1}^{n}\mathbb{E}_k[g_i^k]\right\|^2 + \frac{1}{n^2}\sum_{i=1}^{n}\mathbb{E}_k\left[\left\|g_i^k - \mathbb{E}_k[g_i^k]\right\|^2\right]$$

$$\overset{(6)}{\le} G^2 + \frac{\sigma^2}{n}.$$

Thus, we arrive at the following corollary of Lemma 3:

$$0 \le \mathbb{E}\left[\|x^{k+1} - x^*\|^2\right] \le \|x^0 - x^*\|^2 - 2\sum_{t=0}^{k}\eta_t\mathbb{E}\left[f(x^t) - f(x^*)\right] + 2\sum_{t=0}^{k}\eta_t^2\left(G^2 + \frac{\sigma^2}{n} + \frac{\varepsilon^2}{4n}\right).$$

Furthermore, by convexity of $f$ we have

$$f(\hat{x}^k) - f(x^*) \le \frac{1}{\sum_{t=0}^{k}\eta_t}\sum_{t=0}^{k}\eta_t(f(x^t) - f(x^*)). \tag{12}$$

Plugging it back, rearranging the terms and dropping $\mathbb{E}\left[\|x^{k+1} - x^*\|^2\right]$ gives the result. $\quad\square$

### B.1.5 PROOF OF PROPOSITION 1

*Proof.* Fix any $i$. By Young's inequality and independence of $\xi_1^k, \ldots, \xi_n^k$ we have

$$
\begin{aligned}
\mathbb{E}\left[\left\|\frac{1}{n}\sum_{i=1}^n g_i^k\right\|^2\right] &= \mathbb{E}\left[\left\|\frac{1}{n}\sum_{i=1}^n \nabla f_i(x^k;\xi_i^k)\right\|^2\right] \\
&\leq 2\mathbb{E}\left[\left\|\frac{1}{n}\sum_{i=1}^n \nabla f_i(x^*;\xi_i^k)\right\|^2\right] + 2\mathbb{E}\left[\left\|\frac{1}{n}\sum_{i=1}^n \left(\nabla f_i(x^k;\xi_i^k) - \nabla f_i(x^*;\xi_i^k)\right)\right\|^2\right] \\
&= \frac{2}{n^2}\sum_{i=1}^n \mathbb{E}\left[\|\nabla f_i(x^*;\xi_i^k)\|^2\right] + 2\mathbb{E}\left[\left\|\frac{1}{n}\sum_{i=1}^n \left(\nabla f_i(x^k;\xi_i^k) - \nabla f_i(x^*;\xi_i^k)\right)\right\|^2\right].
\end{aligned}
$$

Substituting the definition of $\sigma_*^2$ and applying Jensen's inequality, we derive

$$
\mathbb{E}\left[\left\|\frac{1}{n}\sum_{i=1}^n g_i^k\right\|^2\right] \leq \frac{\sigma_*^2}{n} + \frac{2}{n}\sum_{i=1}^n \mathbb{E}\left[\|\nabla f_i(x^k;\xi_i^k) - \nabla f_i(x^*;\xi_i^k)\|^2\right].
$$

By our assumption, $f_i(\cdot;\xi)$ is convex and has $L_i$-Lipschitz gradient, so we can use Equation (2.1.7) in Theorem 2.1.5 in Nesterov (2013):

$$
\begin{aligned}
2\mathbb{E}\left[\|\nabla f_i(x^k;\xi_i^k) - \nabla f_i(x^*;\xi_i^k)\|^2\right] &\leq 4L_i\mathbb{E}\left[f_i(x^k;\xi_i^k) - f_i(x^*;\xi_i^k) - \langle\nabla f_i(x^*;\xi_i^k), x^k - x^*\rangle\right] \\
&= 4L_i\mathbb{E}\left[f_i(x^k) - f_i(x^*) - \langle\nabla f_i(x^*), x^k - x^*\rangle\right] \\
&\leq \mathcal{L}\mathbb{E}\left[f_i(x^k) - f_i(x^*) - \langle\nabla f_i(x^*), x^k - x^*\rangle\right].
\end{aligned}
$$

Taking the average over $i = 1, \ldots, n$ and noticing $\sum_{i=1}^n \nabla f_i(x^*) = 0$ yields

$$
\begin{aligned}
\mathbb{E}\left[\left\|\frac{1}{n}\sum_{i=1}^n g_i^k\right\|^2\right] &\leq \frac{\sigma_*^2}{n} + \frac{\mathcal{L}}{n}\sum_{i=1}^n \mathbb{E}\left[f_i(x^k) - f_i(x^*) - \langle\nabla f_i(x^*), x^k - x^*\rangle\right] \\
&= \frac{\sigma_*^2}{n} + \mathcal{L}\mathbb{E}\left[f(x^k) - f(x^*)\right],
\end{aligned}
$$

which is exactly our claim. $\qquad\square$

### B.1.6 PROOF OF THEOREM 2

*Proof.* The proof is almost identical to that of Theorem 1, but now we directly use Assumption 3 and plug it in inside Lemma 3 to get

$$
\begin{aligned}
\mathbb{E}\left[\|x^{k+1} - x^*\|^2\right] &\leq \|x^0 - x^*\|^2 - 2\sum_{t=0}^k \eta_t\mathbb{E}\left[f(x^t) - f(x^*)\right] + 2\sum_{t=0}^k \eta_t^2\mathbb{E}\left[\left\|\frac{1}{n}\sum_{i=1}^n g_i^t\right\|^2\right] + \frac{\varepsilon^2}{2n}\sum_{t=1}^k \eta_t^2 \\
&\overset{(7)}{\leq} \|x^0 - x^*\|^2 - \sum_{t=0}^k 2\eta_t(1 - \eta_t\mathcal{L})\mathbb{E}\left[f(x^t) - f(x^*)\right] + 2\left(\frac{\sigma_*^2}{n} + \frac{\varepsilon^2}{4n}\right)\sum_{t=1}^k \eta_t^2 \\
&\leq \|x^0 - x^*\|^2 - \sum_{t=0}^k \eta_t\mathbb{E}\left[f(x^t) - f(x^*)\right] + 2\left(\frac{\sigma_*^2}{n} + \frac{\varepsilon^2}{4n}\right)\sum_{t=1}^k \eta_t^2.
\end{aligned}
$$

Rearranging this inequality yields

$$
\begin{aligned}
\sum_{t=0}^k \eta_t\mathbb{E}\left[f(x^t) - f(x^*)\right] &\leq \|x^0 - x^*\|^2 - \mathbb{E}\left[\|x^{k+1} - x^*\|^2\right] + 2\left(\frac{\sigma_*^2}{n} + \frac{\varepsilon^2}{4n}\right)\sum_{t=1}^k \eta_t^2 \\
&\leq \|x^0 - x^*\|^2 + 2\left(\frac{\sigma_*^2}{n} + \frac{\varepsilon^2}{4n}\right).
\end{aligned}
$$

To finish the proof, it remains to upper bound $f(\hat{x}^k)$ using convexity the same way as it was done in Equation (12). $\qquad\square$

### B.1.7 PROOF OF THEOREM 3

*Proof.* By $L$-smoothness of $f$ we have

$$\mathbb{E}_k[f(x^{k+1})] \leq f(x^k) + \mathbb{E}_k[\langle \nabla f(x^k), x^{k+1} - x^k \rangle] + \frac{L}{2}\mathbb{E}_k[\|x^{k+1} - x^k\|^2]$$

$$= f(x^k) - \frac{\eta_k}{n}\sum_{i=1}^n \mathbb{E}_k[\langle \nabla f(x^k), Q(g_i^k) \rangle] + \frac{L}{2}\mathbb{E}_k[\|x^{k+1} - x^k\|^2]$$

$$\overset{(3)}{=} f(x^k) - \eta_k\|\nabla f(x^k)\|^2 + \frac{L}{2}\mathbb{E}_k[\|x^{k+1} - x^k\|^2]$$

$$= f(x^k) - \eta_k\|\nabla f(x^k)\|^2 + L\mathbb{E}_k[\|x^{k+1} - x^k\|^2] - \frac{L}{2}\mathbb{E}_k[\|x^{k+1} - x^k\|^2]$$

$$= f(x^k) - \eta_k\|\nabla f(x^k)\|^2 + \eta_k^2 L\mathbb{E}_k\left[\left\|\frac{1}{n}\sum_{i=1}^n Q(g_i^k)\right\|^2\right] - \frac{L}{2}\mathbb{E}_k[\|x^{k+1} - x^k\|^2].$$

Similarly to Lemma 2, we get a decomposition into SGD and quantization errors:

$$\mathbb{E}_k\left[\left\|\frac{1}{n}\sum_{i=1}^n Q(g_i^k)\right\|^2\right] = \mathbb{E}_k\left[\mathbb{E}_Q\left[\left\|\frac{1}{n}\sum_{i=1}^n Q(g_i^k)\right\|^2\right]\right]$$

$$= \mathbb{E}_k\left[\left\|\frac{1}{n}\sum_{i=1}^n g_i^k\right\|^2\right] + \frac{1}{n^2}\sum_{i=1}^n \mathbb{E}_k[\|Q(g_i^k) - g_i^k\|^2]$$

$$\leq \mathbb{E}_k\left[\left\|\frac{1}{n}\sum_{i=1}^n g_i^k\right\|^2\right] + \frac{1}{4n}\sum_{j=1}^d \frac{1}{\alpha_{k,j}^2}.$$

We proceed with the two terms separately. To begin with, we further decompose the SGD error into its expectation and variance:

$$\mathbb{E}_k\left[\left\|\frac{1}{n}\sum_{i=1}^n g_i^k\right\|^2\right] = \left\|\frac{1}{n}\sum_{i=1}^n \mathbb{E}_k[g_i^k]\right\|^2 + \mathbb{E}_k\left[\left\|\frac{1}{n}\sum_{i=1}^n (g_i^k - \mathbb{E}_k[g_i^k])\right\|^2\right]$$

$$= \|\nabla f(x^k)\|^2 + \frac{1}{n^2}\sum_{i=1}^n \mathbb{E}_k\left[\|g_i^k - \nabla f_i(x^k)\|^2\right]$$

$$\overset{(8)}{\leq} \|\nabla f(x^k)\|^2 + \frac{\sigma^2}{n}.$$

Moving on, we plug it back into the upper bound $\mathbb{E}_k[f(x^{k+1})]$. Assuming $\eta_k \leq \frac{1}{2L}$, we get

$$\mathbb{E}_k\left[f(x^{k+1})\right] \leq f(x^k) - \eta_k(1 - \eta_k L)\|\nabla f(x^k)\|^2 + \eta_k^2 L\frac{\sigma^2}{n} + \frac{L}{4n}\sum_{j=1}^d \frac{\eta_k^2}{\alpha_{k,j}^2} - \frac{L}{2}\mathbb{E}_k[\|x^{k+1} - x^k\|^2]$$

$$\leq f(x^k) - \frac{\eta_k}{2}\|\nabla f(x^k)\|^2 + \eta_k^2 L\frac{\sigma^2}{n} + \frac{L}{4n}\sum_{j=1}^d \frac{\eta_k^2}{\alpha_{k,j}^2} - \frac{L}{2}\mathbb{E}_k[\|x^{k+1} - x^k\|^2].$$

Finally, reusing Equation (11) produces the bound

$$\mathbb{E}\left[f(x^{k+1})\right] \leq f(x^0) - \sum_{t=0}^k \frac{\eta_t}{2}\mathbb{E}\left[\|\nabla f(x^t)\|^2\right] + \frac{\sigma^2}{n}\sum_{t=0}^k \eta_t^2 L + \frac{\varepsilon^2}{4n}\sum_{t=0}^k \eta_t^2 L.$$

Notice that by Assumption 4 $f^{\text{inf}} \leq f(x^{k+1})$, so we have

$$\frac{1}{\sum_{t=0}^k \eta_t}\sum_{t=0}^k \eta_t\mathbb{E}\left[\|\nabla f(x^t)\|^2\right] \leq 2\frac{f(x^0) - f^{\text{inf}} + \left(\frac{\sigma^2}{n} + \frac{\varepsilon^2}{4n}\right)\sum_{t=0}^k \eta_t^2 L}{\sum_{t=0}^k \eta_t}.$$

The left-hand side is equal to $\mathbb{E}\left[\|\nabla f(\hat{x}^k)\|^2\right]$ by definition of $\hat{x}^k$, and we conclude the proof. □

### B.1.8 PROOF OF COROLLARY 2

*Proof.* For the first part, we have

$$\frac{\|x^0 - x^*\|^2 + 2\left(G^2 + \frac{\sigma^2}{n} + \frac{\varepsilon^2}{4n}\right)\sum_{t=0}^{k}\eta_t^2}{2\sum_{t=0}^{k}\eta_t} = \mathcal{O}\left(\frac{1}{\sum_{t=0}^{k}\eta_t}\right) = \mathcal{O}\left(\frac{G + \frac{\sigma+\varepsilon}{\sqrt{n}}}{\sqrt{k}}\right).$$

The other complexities follow similarly. □

### B.1.9 PROOF OF PROPOSITION 2

*Proof.* By definition of $\alpha_k$

$$\sum_{j=1}^{d}\mathbb{E}\left[\frac{\eta_k^2}{\alpha_{k,j}^2}\right] = \eta_k^2\varepsilon^2 + 2n\mathbb{E}\left[r_k\right] = \eta_k^2\varepsilon^2 + 2n(1-\beta)\sum_{t=0}^{k-1}\beta^t\|x^{k-t} - x^{k-t-1}\|^2.$$

□

### B.1.10 PROOF OF PROPOSITION 3

*Proof.* Indeed, we only need to plug in the values of $\alpha_{k,j}$:

$$\sum_{j=1}^{d}\mathbb{E}\left[\frac{\eta_k^2}{\alpha_{k,j}^2}\right] = 2n\mathbb{E}\left[\|x^k - x^{k-1}\|^2\right] \overset{\beta=0}{=} 2n(1-\beta)\sum_{t=0}^{k-1}\beta^t\mathbb{E}\left[\|x^{k-t} - x^{k-t-1}\|^2\right].$$

□

### B.1.11 PROOF OF PROPOSITION 4

*Proof.* Since the $l$-th block has $d_l$ coordinates, we get

$$\sum_{j=1}^{d}\mathbb{E}\left[\frac{\eta_k^2}{\alpha_{k,j}^2}\right] = \sum_{l=1}^{B}d_l\mathbb{E}\left[\frac{\eta_k^2}{\alpha_{k,(l)}^2}\right] = 2n\sum_{l=1}^{B}\mathbb{E}\left[\|(x^k)_l - (x^{k-1})_l\|^2\right] = 2n\mathbb{E}\left[\|x^k - x^{k-1}\|^2\right].$$

□

## B.2 PROOFS FOR INTDIANA

**Assumption 5.** $f_{il}(x)$ has $L_{il}$-Lipschitz gradient. We define $\mathcal{L} \overset{\text{def}}{=} 4\max_{i\in[n]}\max_{l\in[m]}L_{il}$.

**Proposition 5.** Suppose that Assumption 5 holds. Then, we have the following for IntDIANA and any $x \in \mathbb{R}^d$:

$$\frac{1}{mn}\sum_{i=1}^{n}\sum_{l=1}^{m}\|\nabla f_{il}(x) - \nabla f_{il}(x^*)\|^2 \le \frac{\mathcal{L}}{2}(f(x) - f(x^*)) \tag{13}$$

*Proof.* Based on Assumption 5 and Theorem 2.1.5 Nesterov (2013), we have:

$$\|\nabla f_{il}(x) - \nabla f_{il}(x^*)\|^2 \le 2L_{il}\left(f_{il}(x) - f_{il}(x^*) - \langle\nabla f_{il}(x^*), x - x^*\rangle\right)$$

Thus, double averaging leads to:

$$\frac{1}{mn}\sum_{i=1}^{n}\sum_{l=1}^{m}\|\nabla f_{il}(x) - \nabla f_{il}(x^*)\|^2$$

$$\le \frac{2}{mn}\sum_{i=1}^{n}\sum_{l=1}^{m}L_{il}\left(f_{il}(x) - f_{il}(x^*) - \langle\nabla f_{il}(x^*), x - x^*\rangle\right)$$

$$\le 2\max_{i}\max_{l}L_{il}\left(f(x) - f(x^*) - \langle\frac{1}{mn}\sum_{i=1}^{n}\sum_{l=1}^{m}\nabla f_{il}(x^*), x - x^*\rangle\right)$$

Considering that $\frac{1}{mn}\sum_{i=1}^{n}\sum_{l=1}^{m}\nabla f_{il}(x^*) = 0$ and defining $4\max_{i\in[n]}\max_{l\in[m]}L_{il}$ leads to the claim in the proposition. □

**Lemma 4.** For IntDIANA (Algorithm 3) and $g^k \stackrel{\text{def}}{=} \frac{1}{n} \sum_{i=1}^{n} (h_i^k + Q(g_i^k))$, we have $\mathbb{E}_k [g^k] = \nabla f(x^k)$ and:

$$\mathbb{E}_k \left[ \|g^k\|^2 \right] \leq \frac{1}{4n} \sum_{j=1}^{d} \frac{1}{\alpha_{k,j}^2} + \mathbb{E}_k \left[ \left\| \frac{1}{n} \sum_{i=1}^{n} g_i^k \right\|^2 \right]. \tag{14}$$

*Proof.* By definition, $g^k = \frac{1}{n} \sum_{i=1}^{n} (h_i^k + Q(g_i^k))$, so

$$\mathbb{E}_k [g^k] = \frac{1}{n} \sum_{i=1}^{n} h_i^k + \frac{1}{n} \sum_{i=1}^{n} \mathbb{E}_k [Q(g_i^k)] \stackrel{(3)}{=} \frac{1}{n} \sum_{i=1}^{n} h_i^k + \frac{1}{n} \sum_{i=1}^{n} \mathbb{E}_k [g_i^k] - \frac{1}{n} \sum_{i=1}^{n} h_i^k = \nabla f(x^k).$$

Thus, we have shown that $g^k$ is an unbiased estimate of $\nabla f(x^k)$. Let us proceed with the second moment of $g^k$:

$$\mathbb{E}_k \left[ \|g^k\|^2 \right] = \mathbb{E}_k \left[ \left\| \frac{1}{n} \sum_{i=1}^{n} \left( \frac{1}{\alpha_k} \circ \mathcal{I}nt(\alpha_k \circ (g_i^k - h_i^k)) - (g_i^k - h_i^k) + g_i^k \right) \right\|^2 \right]$$

$$\stackrel{(3)}{=} \mathbb{E}_k \left[ \left\| \frac{1}{n} \sum_{i=1}^{n} \left( \frac{1}{\alpha_k} \circ \mathcal{I}nt(\alpha_k \circ (g_i^k - h_i^k)) - (g_i^k - h_i^k) \right) \right\|^2 \right] + \mathbb{E}_k \left[ \left\| \frac{1}{n} \sum_{i=1}^{n} g_i^k \right\|^2 \right]$$

$$= \frac{1}{n^2} \sum_{i=1}^{n} \mathbb{E}_k \left[ \left\| \frac{1}{\alpha_k} \circ \mathcal{I}nt(\alpha_k \circ (g_i^k - h_i^k)) - (g_i^k - h_i^k) \right\|^2 \right] + \mathbb{E}_k \left[ \left\| \frac{1}{n} \sum_{i=1}^{n} g_i^k \right\|^2 \right]$$

$$\stackrel{(4)}{\leq} \frac{1}{4n} \sum_{j=1}^{d} \frac{1}{\alpha_{k,j}^2} + \mathbb{E}_k \left[ \left\| \frac{1}{n} \sum_{i=1}^{n} g_i^k \right\|^2 \right].$$

$\square$

**Lemma 5.** If L-SVRG estimator $g_i^k = \nabla f_{il}(x^k; \xi_i^k) - \nabla f_{il}(w_i^k; \xi_i^k) + u_i^k$ is used in IntDIANA, we have $\mathbb{E}_k [g_i^k] = \nabla f_i(x^k)$ and

$$\mathbb{E}_k \left[ \left\| \frac{1}{n} \sum_{i=1}^{n} g_i^k \right\|^2 \right] \leq \left( 2L + \frac{\mathcal{L}}{n} \right) \left( f(x^k) - f(x^*) \right) + \frac{2}{n} \sigma_1^k, \tag{15}$$

$$\mathbb{E}_k \left[ \sigma_1^{k+1} \right] \leq (1 - p)\sigma_1^k + \frac{p\mathcal{L}}{2} \left( f(x^k) - f(x^*) \right), \tag{16}$$

where $\sigma_1^k = \frac{1}{mn} \sum_{i=1}^{n} \sum_{l=1}^{m} \|\nabla f_{il}(w_i^k) - \nabla f_{il}(x^*)\|^2$.

*Proof.* Recall that $\mathbb{E} \left[ \|X - \mathbb{E}[X]\|^2 \right] \leq \mathbb{E} \left[ \|X\|^2 \right]$ for any random variable $X$. For the L-SVRG estimator $g_i^k = \nabla f_{il}(x^k; \xi_i^k) - \nabla f_{il}(w_i^k; \xi_i^k) + u_i^k$, we have:

$$\mathbb{E}_k \left[ \left\| \frac{1}{n} \sum_{i=1}^{n} g_i^k \right\|^2 \right] = \left\| \frac{1}{n} \sum_{i=1}^{n} \nabla f_i(x^k) \right\|^2 + \mathbb{E}_k \left[ \left\| \frac{1}{n} \sum_{i=1}^{n} (g_i^k - \nabla f_i(x^k)) \right\|^2 \right]$$

$$\leq 2L \left( f(x^k) - f(x^*) \right) + \frac{1}{n^2} \sum_{i=1}^{n} \frac{1}{m} \sum_{l'=1}^{m} \left\| \nabla f_{il}(x^k) - \nabla f_{il}(w_i^k) - \frac{1}{m} \sum_{l'=1}^{m} (\nabla f_{il'}(x^k) - \nabla f_{il'}(w_i^k)) \right\|^2$$

$$\leq 2L \left( f(x^k) - f(x^*) \right) + \frac{1}{n^2} \sum_{i=1}^{n} \frac{1}{m} \sum_{l'=1}^{m} \left\| \nabla f_{il}(x^k) - \nabla f_{il}(w_i^k) \right\|^2$$

$$\leq 2L \left( f(x^k) - f(x^*) \right) + \frac{2}{n} \frac{1}{mn} \sum_{i=1}^{n} \sum_{l=1}^{m} \left\| \nabla f_{il}(x^k) - \nabla f_{il}(x^*) \right\|^2 + \frac{2}{n} \frac{1}{mn} \sum_{i=1}^{n} \sum_{l=1}^{m} \left\| \nabla f_{il}(w_i^k) - \nabla f_{il}(x^*) \right\|^2$$

$$\stackrel{(13)}{\leq} \left( 2L + \frac{\mathcal{L}}{n} \right) \left( f(x^k) - f(x^*) \right) + \frac{2}{n} \sigma_1^k,$$

where $\sigma_1^k = \frac{1}{mn} \sum_{i=1}^n \sum_{l=1}^m \|\nabla f_{il}(w_i^k) - f_{il}(x^*)\|^2$. Based on the update of control sequence in L-SVRG, we have:

$$\mathbb{E}_k \left[ \sigma_1^{k+1} \right] = \frac{1-p}{mn} \sum_{i=1}^n \sum_{l=1}^m \left\| \nabla f_{il}(w_i^k) - f_{il}(x^*) \right\|^2 + \frac{p}{mn} \sum_{i=1}^n \sum_{l=1}^m \left\| \nabla f_{il}(x^k) - f_{il}(x^*) \right\|^2$$

$$\overset{(13)}{\leq} (1-p)\sigma_1^k + \frac{p\mathcal{L}}{2} \left( f(x^k) - f(x^*) \right).$$

$\square$

**Lemma 6.** Define $\sigma_2^k \overset{\text{def}}{=} \frac{1}{n} \sum_{i=1}^n \|h_i^k - \nabla f_i(x^*)\|^2$. For IntDIANA algorithm, we have:

$$\mathbb{E}_k \left[ \sigma_2^{k+1} \right] \leq \frac{1}{n} \sum_{i=1}^n \mathbb{E}_k \left[ \|g_i^k - \nabla f_i(x^*)\|^2 \right] + \sum_{j=1}^d \frac{1}{\alpha_{k,j}^2}, \tag{17}$$

For the full gradient, we have $\frac{1}{n} \sum_{i=1}^n \mathbb{E}_k \left[ \|g_i^k - \nabla f_i(x^*)\|^2 \right] \leq \frac{\mathcal{L}}{2}(f(x^k) - f(x^*))$. For the L-SVRG estimator, we have $\frac{1}{n} \sum_{i=1}^n \mathbb{E}_k \left[ \|g_i^k - \nabla f_i(x^*)\|^2 \right] \leq 4\sigma_1^k + 3\mathcal{L}(f(x^k) - f(x^*))$.

*Proof.* We define $\sigma_2^k \overset{\text{def}}{=} \frac{1}{n} \sum_{i=1}^n \|h_i^k - \nabla f_i(x^*)\|^2$. Consider the step $h_i^{k+1} = h_i^k + Q(g_i^k)$, where $Q(g_i^k) = \frac{1}{\alpha_k} \circ \mathcal{I}nt(\alpha_k \circ (g_i^k - h_i^k))$. Note that $\mathbb{E}_k \left[ \langle g_i^k - h_i^k, g_i^k - 2\nabla f_i(x^*) + h_i^k \rangle \right] = \mathbb{E}_k \left[ \|g_i^k - \nabla f_i(x^*)\|^2 - \|h_i^k - \nabla f_i(x^*)\|^2 \right]$, which explains the last equality below:

$$\mathbb{E}_k \left[ \sigma_2^{k+1} \right] = \mathbb{E}_k \left[ \frac{1}{n} \sum_{i=1}^n \|h_i^k - \nabla f_i(x^*) + Q(g_i^k)\|^2 \right]$$

$$= \sigma_2^k + \frac{1}{n} \sum_{i=1}^n \mathbb{E}_k \left[ \left\| \frac{1}{\alpha_k} \circ \mathcal{I}nt \left( \alpha_k \circ (g_i^k - h_i^k) \right) \right\|^2 \right] + 2\frac{1}{n} \sum_{i=1}^n \mathbb{E}_k \left[ \langle Q(g_i^k), h_i^k - \nabla f_i(x^*) \rangle \right]$$

$$\overset{(4)}{=} \sigma_2^k + \frac{1}{n} \sum_{i=1}^n \mathbb{E}_k \left[ \|g_i^k - h_i^k\|^2 \right] + 2\frac{1}{n} \sum_{i=1}^n \langle g_i^k - h_i^k, h_i^k - \nabla f_i(x^*) \rangle + \sum_{j=1}^d \frac{1}{\alpha_{k,j}^2}$$

$$\leq \sigma_2^k + \frac{1}{n} \sum_{i=1}^n \mathbb{E}_k \left[ \langle g_i^k - h_i^k, g_i^k - 2\nabla f_i(x^*) + h_i^k \rangle \right] + \sum_{j=1}^d \frac{1}{\alpha_{k,j}^2}$$

$$= \frac{1}{n} \sum_{i=1}^n \mathbb{E}_k \left[ \|g_i^k - \nabla f_i(x^*)\|^2 \right] + \sum_{j=1}^d \frac{1}{\alpha_{k,j}^2}.$$

For the full gradient $g_i^k = \nabla f_i(x^k)$, we have:

$$\frac{1}{n} \sum_{i=1}^n \mathbb{E}_k \left[ \|g_i^k - \nabla f_i(x^*)\|^2 \right] = \frac{1}{n} \sum_{i=1}^n \mathbb{E}_k \left[ \|\nabla f_i(x^k) - \nabla f_i(x^*)\|^2 \right] \leq \frac{\mathcal{L}}{2}(f(x^k) - f(x^*)).$$

For the L-SVRG estimator, we have by Young's inequality:

$$\frac{1}{n} \sum_{i=1}^n \mathbb{E}_k \left[ \|g_i^k - \nabla f_i(x^*)\|^2 \right] \leq \frac{2}{n} \sum_{i=1}^n \mathbb{E}_k \left[ \|g_i^k - \nabla f_i(x^k)\|^2 \right] + \frac{2}{n} \sum_{i=1}^n \|\nabla f_i(x^k) - \nabla f_i(x^*)\|^2$$

$$\leq \frac{2}{mn} \sum_{i=1}^n \sum_{l=1}^m \|\nabla f_{il}(x^k) - \nabla f_{il}(w_i^k)\|^2 + \mathcal{L}(f(x^k) - f(x^*))$$

$$\leq \frac{4}{mn} \sum_{i=1}^n \sum_{l=1}^m \|\nabla f_{il}(x^k) - \nabla f_{il}(x^*)\|^2 + \frac{4}{mn} \sum_{i=1}^n \sum_{l=1}^m \|\nabla f_{il}(w_i^k) - \nabla f_{il}(x^*)\|^2 + \mathcal{L}(f(x^k) - f(x^*))$$

$$\overset{(13)}{\leq} 4\sigma_1^k + 3\mathcal{L}(f(x^k) - f(x^*)).$$

$\square$

**Lemma 7.** Suppose that Assumption 5 holds. Besides, we assume that $f(\cdot)$ is $\mu$-strongly convex ($\mu \geq 0$). For IntDIANA with adaptive $\alpha_k = \frac{\eta_k \sqrt{d}}{\sqrt{n}\|x^k - x^{k-1}\|}$ and GD gradient estimator, we have:

$$\mathbb{E}_k\left[\|x^{k+1} - x^*\|^2\right] + \mathbb{E}_k\left[\|x^{k+1} - x^k\|^2\right]$$
$$\leq (1 - \eta_k\mu)\|x^k - x^*\|^2 + \frac{1}{2}\|x^k - x^{k-1}\|^2 - 2\eta_k(1 - 2\eta_k L)(f(x^k) - f(x^*)),$$
$$\mathbb{E}_k\left[\sigma_2^{k+1}\right] \leq \frac{\mathcal{L}}{2}(f(x^k) - f(x^*)) + n\|x^k - x^{k-1}\|^2.$$

For IntDIANA with adaptive $\alpha_k$ and L-SVRG gradient estimator, we have:

$$\mathbb{E}_k\left[\|x^{k+1} - x^*\|^2\right] + \mathbb{E}_k\left[\|x^{k+1} - x^k\|^2\right]$$
$$\leq (1 - \eta_k\mu)\|x^k - x^*\|^2 + \frac{1}{2}\|x^k - x^{k-1}\|^2 - 2\eta_k\left(1 - 2\eta_k\left(L + \frac{\mathcal{L}}{2n}\right)\right)(f(x^k) - f(x^*)) + \frac{4\eta_k^2}{n}\sigma_1^k,$$
$$\mathbb{E}_k\left[\sigma_1^{k+1}\right] \leq (1 - p)\sigma_1^k + \frac{p\mathcal{L}}{2}(f(x^k) - f(x^*)),$$
$$\mathbb{E}_k\left[\sigma_2^{k+1}\right] \leq 4\sigma_1^k + 3\mathcal{L}(f(x^k) - f(x^*)) + n\|x^k - x^{k-1}\|^2.$$

*Proof.* By $\mu$-strong convexity, we have:

$$\mathbb{E}_k\left[-2\frac{\eta_k}{n}\sum_{i=1}^{n}\langle Q(g_i^k), x^k - x^*\rangle\right] \overset{(3)}{=} -2\frac{\eta_k}{n}\sum_{i=1}^{n}\langle \mathbb{E}_k[g_i^k], x^k - x^*\rangle$$
$$\leq -2\eta_k(f(x^k) - f(x^*)) - \eta_k\mu\|x^k - x^*\|^2.$$

Besides, $\|x^{k+1} - x^k\|^2 = 2\eta_k^2\|g^k\|^2 - \|x^{k+1} - x^k\|^2$, so

$$\mathbb{E}_k\left[\|x^{k+1} - x^*\|^2\right] + \mathbb{E}_k\left[\|x^{k+1} - x^k\|^2\right]$$
$$= (1 - \eta_k\mu)\|x^k - x^*\|^2 - 2\eta_k(f(x^k) - f(x^*)) + 2\eta_k^2\mathbb{E}_k\left[\|g^k\|^2\right]$$
$$\overset{(14)}{\leq} (1 - \eta_k\mu)\|x^k - x^*\|^2 - 2\eta_k(f(x^k) - f(x^*)) + 2\eta_k^2\mathbb{E}_k\left[\left\|\frac{1}{n}\sum_{i=1}^{n}g_i^k\right\|^2\right] + \frac{1}{2n}\sum_{j=1}^{d}\frac{\eta_k^2}{\alpha_{k,j}^2}$$

Applying Proposition 3 to the obtained bound results in the following recursion

$$\mathbb{E}_k\left[\|x^{k+1} - x^*\|^2\right] + \mathbb{E}_k\left[\|x^{k+1} - x^k\|^2\right]$$
$$\leq (1 - \eta_k\mu)\|x^k - x^*\|^2 + \frac{1}{2}\mathbb{E}_k\left[\|x^k - x^{k-1}\|^2\right] - 2\eta_k(f(x^k) - f(x^*)) + 2\eta_k^2\mathbb{E}_k\left[\left\|\frac{1}{n}\sum_{i=1}^{n}g_i^k\right\|^2\right].$$

With the GD estimator, the produced bound simplifies to

$$\mathbb{E}_k\left[\|x^{k+1} - x^*\|^2\right] + \mathbb{E}_k\left[\|x^{k+1} - x^k\|^2\right]$$
$$\leq (1 - \eta_k\mu)\|x^k - x^*\|^2 + \frac{1}{2}\|x^k - x^{k-1}\|^2 - 2\eta_k(1 - 2\eta_k L)(f(x^k) - f(x^*)).$$

Based on Lemma 6, the following is satisfied for IntDIANA with GD estimator and adaptive $\alpha_k = \frac{\eta_k \sqrt{d}}{\sqrt{n}\|x^k - x^{k-1}\|}$:

$$\mathbb{E}_k\left[\sigma_2^{k+1}\right] \leq \frac{\mathcal{L}}{2}(f(x^k) - f(x^*)) + n\|x^k - x^{k-1}\|^2.$$

In turn, Lemma 5 gives for L-SVRG estimator $\mathbb{E}_k\left[\left\|\frac{1}{n}\sum_{i=1}^{n}g_i^k\right\|^2\right] \leq \left(2L + \frac{\mathcal{L}}{n}\right)(f(x^k) - f(x^*)) + \frac{2}{n}\sigma_1^k$, so we can derive that

$$\mathbb{E}_k\left[\|x^{k+1} - x^*\|^2\right] + \mathbb{E}_k\left[\|x^{k+1} - x^k\|^2\right]$$
$$\leq (1 - \eta_k\mu)\|x^k - x^*\|^2 + \frac{1}{2}\|x^k - x^{k-1}\|^2 - 2\eta_k\left(1 - 2\eta_k\left(L + \frac{\mathcal{L}}{2n}\right)\right)(f(x^k) - f(x^*)) + \frac{4\eta_k^2}{n}\sigma_1^k.$$

Let us now combine Equation (16) and Lemma 6:

$$\mathbb{E}_k\left[\sigma_1^{k+1}\right] \le (1-p)\sigma_1^k + \frac{p\mathcal{L}}{2}(f(x^k) - f(x^*)),$$

$$\mathbb{E}_k\left[\sigma_2^{k+1}\right] \le 4\sigma_1^k + 3\mathcal{L}(f(x^k) - f(x^*)) + n\|x^k - x^{k-1}\|^2.$$

$\square$

**Lemma 8.** We define the Lyapunov function as $\Psi^k \stackrel{\text{def}}{=} \|x^k - x^*\|^2 + \|x^k - x^{k-1}\|^2 + c_1\eta_k^2\sigma_1^k + c_2\eta_k^2\sigma_2^k$. Assume that the conditions of Lemma 7 hold. If $\mu > 0$, we have:

$$\mathbb{E}\left[\Psi^{k+1}\right] \le \theta\mathbb{E}\left[\Psi^k\right],$$

where $\theta \stackrel{\text{def}}{=} \max\left\{(1 - \eta_k\mu), \frac{3}{4}\right\}$, $c_1 = 0$, $c_2 = \frac{L^2}{4n}$ and $\eta_k \le \frac{1}{2(L + \frac{\mathcal{L}}{32n})}$ for IntDIANA with GD estimator. Alternatively, for IntDIANA with L-SVRG estimator, we have $\theta \stackrel{\text{def}}{=} \max\left\{(1 - \eta_k\mu), \frac{3}{4}, \left(1 - \frac{3}{8m}\right)\right\}$ and set $c_1 = \frac{8m}{n}$, $c_2 = \frac{L^2}{4n}$, $p = \frac{1}{m}$, and $\eta_k \le \frac{1}{2(L + 2\mathcal{L}/n)}$. If $\mu = 0$, we have

$$\eta_k\mathbb{E}\left[f(x^k) - f(x^*)\right] \le \mathbb{E}\left[\Psi^k\right] - \mathbb{E}\left[\Psi^{k+1}\right],$$

where $\eta_k \le \frac{1}{4(L + \frac{\mathcal{L}}{32n})}$ for the GD variant and $\eta_k \le \frac{1}{4(L + 2\mathcal{L}/n)}$ for the L-SVRG variant.

*Proof.* We define the Lyapunov function as $\Psi^k \stackrel{\text{def}}{=} \|x^k - x^*\|^2 + \|x^k - x^{k-1}\|^2 + c_1\eta_k^2\sigma_1^k + c_2\eta_k^2\sigma_2^k$. For IntDIANA with GD estimator, we can set $c_1 = 0$ and derive the following inequality from Lemma 7 and $\eta_{k+1} \le \eta_k$:

$$\mathbb{E}\left[\Psi^{k+1}\right] \le (1 - \eta_k\mu)\mathbb{E}\left[\|x^k - x^*\|^2\right] + \left(\frac{1}{2} + c_2\eta_k^2 n\right)\mathbb{E}\left[\|x^k - x^{k-1}\|^2\right]$$

$$- 2\eta_k\left(1 - 2\eta_k\left(L + \frac{c_2\mathcal{L}}{8}\right)\right)\mathbb{E}\left[f(x^k) - f(x^*)\right].$$

We first consider $\mu > 0$ case. Let $c_2 = \frac{L^2}{4n}$, and $\eta_k \le \frac{1}{2(L + \frac{\mathcal{L}}{32n})}$. We have $\mathbb{E}\left[\Psi^{k+1}\right] \le \max\left\{(1 - \eta_k\mu), \frac{3}{4}\right\}\mathbb{E}\left[\Psi^k\right]$.

For IntDIANA with L-SVRG estimator, we have the following based on Lemma 7:

$$\mathbb{E}\left[\Psi^{k+1}\right] \le (1 - \eta_k\mu)\mathbb{E}\left[\|x^k - x^*\|^2\right] + \left(\frac{1}{2} + c_2\eta_k^2 n\right)\mathbb{E}\left[\|x^k - x^{k-1}\|^2\right]$$

$$+ \eta_k^2\left(\frac{4}{n} + 4c_2 + (1-p)c_1\right)\mathbb{E}\left[\sigma_1^k\right]$$

$$- 2\eta_k\left(1 - 2\eta_k\left(L + \frac{\mathcal{L}}{2n} + \frac{pc_1\mathcal{L}}{8} + \frac{3c_2\mathcal{L}}{4}\right)\right)\mathbb{E}\left[f(x^k) - f(x^*)\right]$$

Let $c_1 = \frac{8m}{n}$, $c_2 = \frac{L^2}{4n}$, $p = \frac{1}{m}$, and $\eta_k \le \frac{1}{2(L + 2\mathcal{L}/n)}$. Plugging these values into the recursion, we get $\mathbb{E}\left[\Psi^{k+1}\right] \le \max\left\{(1 - \eta_k\mu), \frac{3}{4}, \left(1 - \frac{3}{8m}\right)\right\}\mathbb{E}\left[\Psi^k\right]$.

If $\mu = 0$, we instead let $\eta_k \le \frac{1}{4(L + \frac{\mathcal{L}}{32n})}$ for the GD variant and $\eta_k \le \frac{1}{4(L + 2\mathcal{L}/n)}$ for the L-SVRG variant to obtain from the same recursions:

$$\eta_k\mathbb{E}\left[f(x^k) - f(x^*)\right] \le \mathbb{E}\left[\Psi^k\right] - \mathbb{E}\left[\Psi^{k+1}\right].$$

$\square$

## C  DETAILS AND ADDITIONAL RESULTS OF EXPERIMENTS

### C.1  MORE DETAILS

Here we provide more details of our experimental setting. We use the learning rate scaling technique (Goyal et al., 2017; Vogels et al., 2019) with 5 warm-up epochs. As suggested in previous works

(Vogels et al., 2019; Alistarh et al., 2017; Horváth et al., 2019), we tune the initial single-worker learning rate on the full-precision SGD and then apply it to PowerSGD, QSGD, and NatSGD.

For the task of training ResNet18 on the CIFAR-10 dataset, we utilize momentum $\beta = 0.9$ and weight decay with factor $10^{-4}$ (except the Batchnorm parameters) for all algorithms. All algorithms run for 300 epochs. The learning rate decays by 10 times at epoch 150 and 250. The initial learning rate is tuned in the range {0.05, 0.1, 0.2, 0.5} and we choose 0.1.

For the task of training a 3-layer LSTM, all algorithms run for 90 epochs. We set the size of word embeddings to 650, the sequence length to 30, the number of hidden units per layer to 650, and the dropout rate to 0.4. Besides, we tie the word embedding and softmax weights. We tune the initial learning rate in the range of {0.6, 1.25, 2.5, 5} and we choose 1.25. For both tasks, we report the results based on 3 repetitions with random seeds {0, 1, 2}. To measure the time of computation, communication, and compression/decompression of the algorithms, we use the timer (a Python context manager) implemented in the PowerSGD code[6].

For PowerSGD, we use rank = 2 in the task of training ResNet18 on the CIFAR-10 dataset and rank = 4 in the task of training LSTM on the Wikitext-2 dataset as suggested by Vogels et al. (2019). For QSGD, we use the gradient matrix of each layer as a bucket and set the number of quantization levels to be 64 (6-bit).

## C.2 TOY EXPERIMENT ON TIMINGS

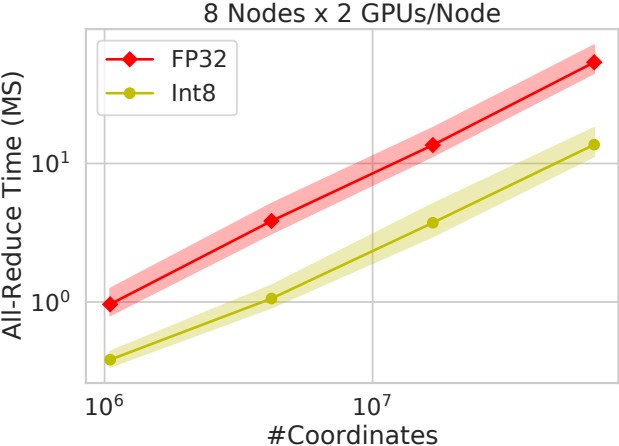

Figure 2: Time of communicating FP32 and Int8 messages based on all-reduce.

Figure 2 shows the different manners of PowerSGD and IntSGD to save the communication time based on the all-reduce compared to full-precision SGD: 1) IntSGD (8-bit) communicates the data with `int8` data type but does not reduce the number of coordinates; 2) PowerSGD does not change the data type but breaks one communication round of a big number of coordinates into three communication rounds of much smaller numbers of coordinates.

## C.3 CONVERGENCE CURVES OF THE DEEP LEARNING TASKS

Please see Figure 3 and Figure 4.

## C.4 SENSITIVITY ANALYSIS OF HYPERPARAMETERS

We analyze the sensitivity of IntSGD to its hyperparameters $\beta$ and $\varepsilon$. As shown in Figure 5, the performance of IntSGD is quite stable across the choices of $\beta \in \{0.0, 0.3, 0.6, 0.9\}$ and $\varepsilon \in \{10^{-4}, 10^{-6}, 10^{-8}\}$ on the two considered tasks. Overall, $\beta = 0.9$ and $\varepsilon = 10^{-8}$ is a good default setting for our IntSGD algorithm.

---

[6] https://github.com/epfml/powersgd

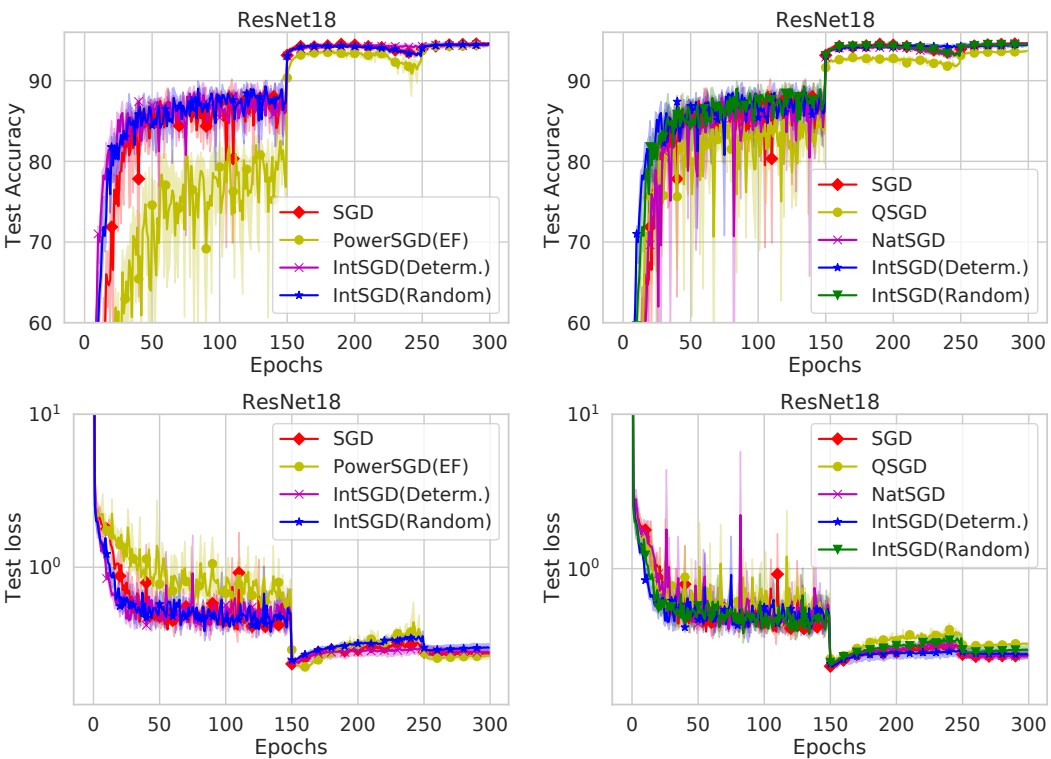

Figure 3: Convergence curves of IntSGD (Random) and IntSGD (Determ.) and the baseline algorithms on the task of training ResNet18 on the CIFAR-10 dataset.

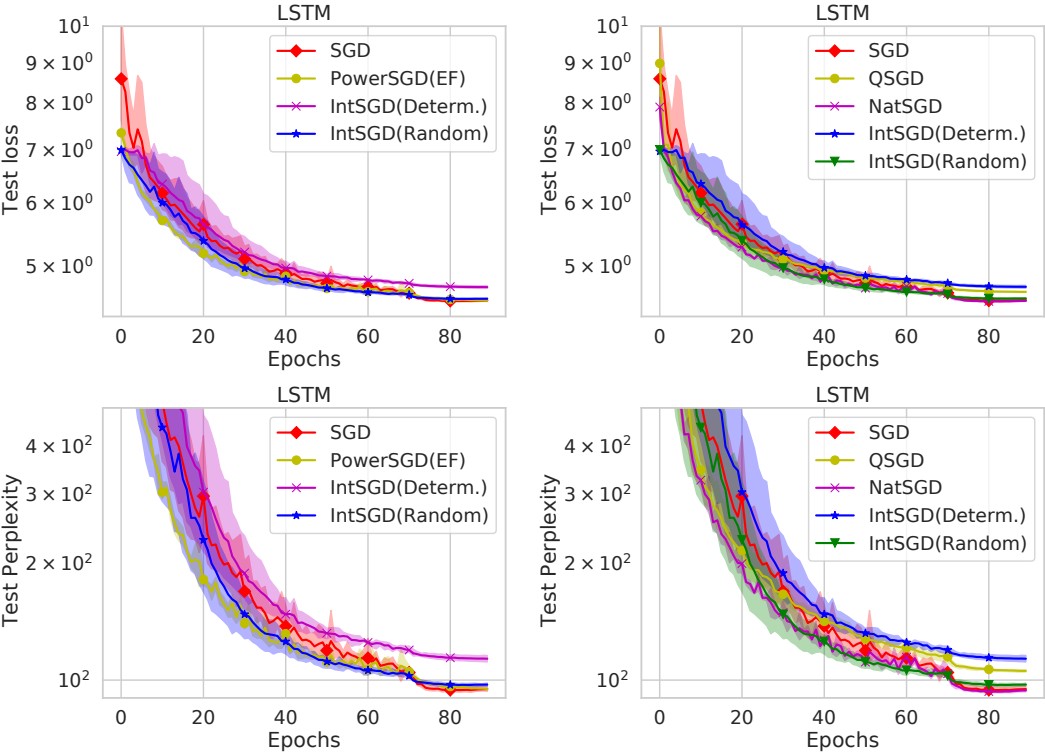

Figure 4: Convergence curves of IntSGD (Random) and IntSGD (Determ.) and the baseline algorithms on the task of training a 3-layer LSTM on the Wikitext-2 dataset.

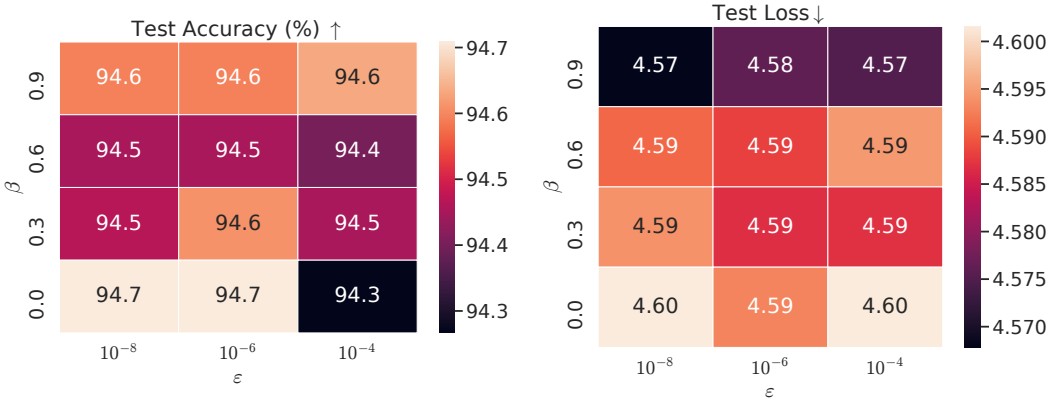

Figure 5: Test accuracy (on the image classification task) and test loss (on the language modeling task) of IntSGD under different hyperparameters $\beta$ and $\varepsilon$. "↑" or "↓" denotes the larger, the better or vice versa.

### C.5 LOGISTIC REGRESSION EXPERIMENT

**Setup:** We run the experiments on the $\ell_2$-regularized logistic regression problem with four datasets (a5a, mushrooms, w8a, real-sim) from the LibSVM repository[7], where

$$f(x) = \frac{1}{n} \sum_{i=1}^{n} f_i(x)$$

and

$$f_i(x) = \frac{1}{m} \sum_{l=1}^{m} \log(1 + \exp(-\mathbf{A}_{i,l}^{\top} x b_{i,l})) + \frac{\lambda_2}{2} \|x\|^2,$$

and $x \in \mathbb{R}^d$, $\lambda_2$ is chosen proportionally to $\frac{1}{mn}$ and $\mathbf{A}_{i,l} \in \mathbb{R}^d$, $b_{i,l} \in \{-1, 1\}$ are the feature and label of $l$-th data point on the $i$-th worker. The experiments are performed on a machine with 24 Intel(R) Xeon(R) Gold 6246 CPU @ 3.30GHz cores, where 12 cores are connected to a socket (there are two sockets in total). All experiments use 12 cpu cores and each core is utilized as a worker. The communications are implemented based on the MPI4PY library Dalcín et al. (2005). The "optimum" $x^*$ is obtained by running GD with the whole data using one cpu core until there are 5000 iterations or $\|\nabla f(x)\|^2 \leq 10^{-30}$.

Table 4: Information of the experiments on $\ell_2$-regularized logistic regression.

| Dataset | #Instances $N$ | Dimension $d$ | $\lambda_2$ |
|---------|----------------|---------------|-------------|
| a5a | 6414 | 123 | $5 \times 10^{-4}$ |
| mushrooms | 8124 | 112 | $6 \times 10^{-4}$ |
| w8a | 49749 | 300 | $10^{-4}$ |
| real-sim | 72309 | 20958 | $5 \times 10^{-5}$ |

The whole dataset is split according to its original indices into $n$ folds, and each fold is assigned to a local worker, i.e., the data are heterogeneous. There are $m = \lfloor \frac{N}{n} \rfloor$ data points on each worker. For each dataset, we run each algorithm multiples times with 20 random seeds for each worker. For the stochastic algorithms, we randomly sample 5% of the local data as a minibatch (i.e., batch size $\tau = \lfloor \frac{m}{20} \rfloor$) to estimate the stochastic gradient $g_i^k$ on each worker. We set $p = \frac{\tau}{m}$ in VR-IntDIANA.

---

[7]https://www.csie.ntu.edu.tw/ cjlin/libsvmtools/datasets/binary.html

Apart from IntSGD with $g_i^k = \nabla f_i(x^k)$ (IntGD), we also evaluate IntDIANA (Algorithm 3) with the GD or L-SVRG estimator (called IntDIANA and VR-IntDIANA, respectively).

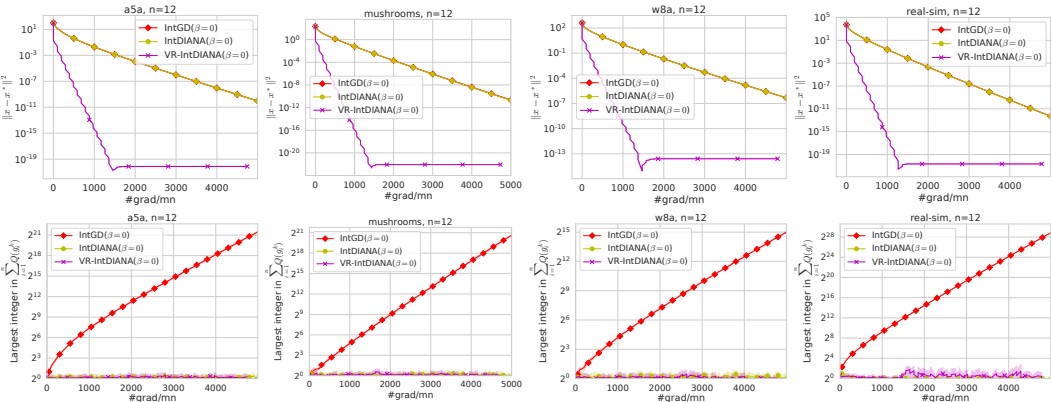

Figure 6: Objective gaps and the max integer in the aggregated vector $\sum_{i=1}^n Q(g_i^k)$.

As shown in Figure 6, IntSGD with $g_i^k = \nabla f_i(x^k)$ (IntGD) suffers from low compression efficiency issue (very large integer in the aggregated vector $\sum_{i=1}^n Q(g_i^k)$) and IntDIANA can solve this issue and only requires less then 3 bits per coordinate in the communication. IntDIANA with SVRG gradient estimator (VR-IntDIANA) further improves IntDIANA with GD estimator in terms of gradient oracles.

