# OpenReview forum: "IntSGD: Adaptive Floatless Compression of Stochastic Gradients"
_ICLR.cc/2022/Conference — ICLR 2022 Spotlight_

### Official Review · Reviewer_GF5a · 2021-10-30

**Correctness:** 3
**Technical Novelty And Significance:** 3
**Empirical Novelty And Significance:** 3
**Recommendation:** 6
**Confidence:** 3

**Main Review:**

The authors find a clever scaling factor that makes the convergence proof work. In the experiments, they show that their method works best on one task, but not as well as PowerSGD on another task.

* Can this approach be extended to popular optimization methods such as Adam?
* In Table 2, it will be useful to have the results for Heuristic SGD as well. How does Heuristic SGD perform without the factor $n$ in the denominator of the scaling factor? That will reduce the quantization error, at the cost of removing the protection against overflow. There is no guard against overflow in IntSGD as well.

Minor:
* In Algorithm 1, line 6, is $\| x^k - x^{k-1} \|^2$ a scalar? Is $r_0 = 0 \in R^d$? That would mean all the entries of $r_k$ are equal at any iteration $k$. Looking at Algorithm 2 in the appendix clarifies this, but the authors should explain this in the main text.
* Page 6: What is the linear speedup with respect to?
* After Assumption 1, the authors claim they show several choices of scaling factors $\alpha$. But they show only one choice (the other choice in the supplement is a special case of the first choice) Please adjust the claim accordingly.
* In Tables 1, 2, why are the communication latencies so different for IntSGD(Determ) and IntSGD (Random)? The discrepancies are too high be explained with the standard deviations reported.
* Please run through a spell-checker to fix the few typos.


**Summary Of The Paper:**

The authors propose a quantized parallel SGD where the gradient coordinates are rounded after scaling: $Q_\alpha(g) = $ round$(\alpha g )/\alpha.$ The scaling factor $\alpha$ determines the quantization error: higher the $\alpha$, lower the quantization error. The authors propose a clever $\alpha$ shared across workers. Such sharing of $\alpha$ allows for in-network aggregation of gradient updates using programmable switches. It also allows for an all-reduce model of parallel SGD which can be much more efficient than the all-gather model. A previous version Heuristic IntSGD uses a scaling factor that is more intuitive (uses all available bits binning the known range), but interestingly its accuracy is suboptimal on some tasks. Other integer rounding methods such as QSGD use a scale $\alpha$ that is not the same across workers and hence it does not allow the all-reduce primitive.



**Summary Of The Review:**

The scaling factor proposed by the authors helps them show convergence of their quantized SGD. Empirically, the authors were able to show that their method beats competitors only one task out of the two considered.

---

> ### Author Response · Authors · 2021-11-23
> **We have revised the paper and ran additional experiments to answer your questions (part II of the response)**
>
> (continued response)
>
> > After Assumption 1, the authors claim they show several choices of scaling factors. But they show only one choice (the other choice in the supplement is a special case of the first choice).
>
> In the updated manuscript, we clarified in the text that one choice is presented in Section 4 and the other choices are given in the supplement. Nevertheless, the choice from Proposition 4 in the supplementary material is **not** a special case of the first choice since it uses block values in alpha, where the first choice is scalar.
>
> > In Tables 1, 2, why are the communication latencies so different for IntSGD(Determ) and IntSGD (Random)? The discrepancies are too high to be explained with the standard deviations reported.
>
> This is a good observation. The discrepancy is about 0.2 ~ 0.7 milliseconds. We suspect that this is due to the compression functions affecting the communication latency since the synchronous all-reduce can only be executed after all workers complete the forward, backward, and compression steps. In a real cluster of nodes, the durations of completing those steps on different workers are very likely to have some discrepancies. Thus, Random and Deterministic compressors might lead to different waiting times, which can be a part of the communication latency we measured. Since the standard deviations in Tables 1, 2 are based on the results for different random seeds, they do not accommodate for the deviation of the cluster communication. We ran an additional test and observed that sometimes the difference between latencies slightly changes, but the comparison of the total time is nevertheless reliable.
>
> > Please run through a spell-checker to fix the few typos.
>
> Thank you for the suggestion. We ran the paper through Grammarly, which suggested 14 typos on top of the ones pointed out by the reviewers. We fixed all of them.

---

> ### Author Response · Authors · 2021-11-23
> **We have revised the paper and ran additional experiments to answer your questions (part I of the response)**
>
> We thank the reviewer for the constructive feedback.
> > Can this approach be extended to popular optimization methods such as Adam?
>
> Do you mean a theoretical extension or an empirical one? From the theoretical perspective, it might be possible to extend our results to Adam by combining our proof with the proof techniques developed in [1]. However, the convergence theory of Adam itself is far from satisfactory, and the theory in [1] requires error feedback and relies on strong assumptions.
> In practice, it is possible to combine our compressor with Adam since our technique gives an unbiased estimate of the gradient. However, prior work [2] shows that the test performance of Adam might be inferior to SGD/SGDm on the tasks we considered (image classification and language modeling). Adam seems to be more popular for training GANs, graph neural networks, and Q-learning with function approximation [2, 3].
>
> [1] Chen et al., “Quantized Adam with Error Feedback”, 2021, ACM Transactions on Intelligent Systems and Technology.
> [2] Wilson et al., “The Marginal Value of Adaptive Gradient Methods in Machine Learning”, NeurIPS 2017.
> [3] Hamilton et al., “Inductive Representation Learning on Large Graphs”, NeurIPS 2017.
>
> > In Table 2, it will be useful to have the results for Heuristic SGD as well.
>
> Thanks for the suggestion! Below we attached the full tables including the results for Heuristic IntSGD. We cannot directly add those results to the updated manuscript due to the 9-page limit.
>
> Table 2: Test accuracy and time breakdown in one iteration (on average) of training ResNet18 on the Cifar10 dataset with 16 workers. All numbers of time are in millisecond (ms).
>
> Algorithm | Test Accuracy (%)|Overhead|Communication|Total Time
> :-----:|:-----:|:-----:|:-----:|:-----:
> SGD (All-gather)  | 94.65 $\pm$ 0.08 | - |  261.29 $\pm$ 0.98 |  338.76 $\pm$ 0.76
> QSGD |  93.69 $\pm$ 0.03 | 129.25 $\pm$ 1.58 | 138.16 $\pm$ 1.29 | 320.49 $\pm$ 2.11
> NatSGD | 94.57 $\pm$ 0.13 |  36.01 $\pm$ 1.30 |  106.27 $\pm$ 1.43 | 197.18 $\pm$ 0.25
> SGD (All-reduce) | 94.67 $\pm$ 0.17| - | 18.48 $\pm$ 0.09 | 74.32 $\pm$ 0.06
> Heuristic IntSGD | $\textcolor{red}{90.96 \pm 1.67}$ | 2.34 $\pm$ 0.04 | 7.16 $\pm$ 0.04 | 64.89 $\pm$ 0.06
> PowerSGD (EF) | 94.33 $\pm$ 0.15 | 7.07 $\pm$ 0.03 | 5.03 $\pm$ 0.07 | 67.08 $\pm$ 0.06
> IntSGD (Determ.) |  94.43 $\pm$ 0.12 | 2.51 $\pm$ 0.04 | 6.92 $\pm$ 0.07 | 64.95 $\pm$ 0.15
> IntSGD (Random) |  94.55 $\pm$ 0.13 | 3.20 $\pm$ 0.02 | 6.21 $\pm$ 0.13 | 65.22 $\pm$ 0.08
>
> Table 3: Test loss and time breakdown in one iteration (on average) of training a 3-layer LSTM on the Wiki-text2 dataset with 16 workers. All numbers of time are in millisecond (ms).
>
> Algorithm | Test Loss |Overhead|Communication|Total Time
> :-----:|:-----:|:-----:|:-----:|:-----:
> SGD (All-gather)  | 4.52 $\pm$ 0.01 | - | 733.07 $\pm$ 1.04 | 796.23 $\pm$ 1.03
> QSGD | 4.63 $\pm$ 0.01 | 43.67 $\pm$ 0.11 | 307.63 $\pm$ 1.16 | 399.10 $\pm$ 1.25
> NatSGD | 4.52 $\pm$ 0.01 | 64.63 $\pm$ 0.12 | 309.87 $\pm$ 1.32 | 422.49 $\pm$ 2.15
> SGD (All-reduce) | 4.54 $\pm$ 0.03 | - | 22.33 $\pm$ 0.02 | 70.46 $\pm$ 0.05
> Heuristic IntSGD | $\textcolor{red}{6.89 \pm 0.10}$ | 2.39 $\pm$ 0.01 | 7.17 $\pm$ 0.01 | 57.79 $\pm$ 0.06
> PowerSGD (EF) | 4.52 $\pm$ 0.01 | 4.22 $\pm$ 0.01 | 2.10 $\pm$ 0.01 | 54.89 $\pm$ 0.02
> IntSGD (Determ.) |  4.70 $\pm$ 0.02 | 3.04 $\pm$ 0.01 | 6.94 $\pm$ 0.05 | 57.93 $\pm$ 0.03
> IntSGD (Random) | 4.54 $\pm$ 0.01 | 4.76 $\pm$ 0.01 | 7.14 $\pm$ 0.04 | 59.99 $\pm$ 0.01
>
> Note that Heuristic IntSGD fails to match the test performance of full precision SGD.
>
> > How does Heuristic SGD perform without the factor n in the denominator of the scaling factor?
>
> To answer your question, we ran an additional ResNet18 experiment (under the same setting) to testify the Heuristic IntSGD without the factor $n$ in the denominator of their scaling factor, i.e., the scaling factor is $n$ times larger. It turns out that the training loss explodes in the first several iterations. The test losses in the first 5 epochs are 41.6, 30.7, 19128238.0, 35468108365824.0, 1095929893486592.0. The reason might be too many overflows. We suspect that some factor between 1 and $n$ might work well for Heuristic IntSGD, but it may lead to less stable convergence and may require more tuning.
>
> > In Algorithm 1, line 6, is |x^k-x^{k-1}|^2 a scalar?
>
> Yes, it is a scalar. We added some clarification in the updated manuscript.
>
> > Page 6: What is the linear speedup with respect to?
>
> When the leading term is $O(1/\sqrt{nk})$, it has linear speedup w.r.t. running the algorithm on a single machine, i.e., more machines lead to a smaller number of iterations to achieve the same level of accuracy. In other words, if we need to run the method for $k$ iterations when $n=1$, we only need $k/n$ iterations when we increase the number of machines.

---

> > ### Comment · Reviewer_GF5a · 2021-11-28
> > **Thank you for the clarifications.**
> >
> > I thank the authors for addressing my concerns satisfactorily. It is surprising that the Heuristic IntSGD is so much worse compared to the proposed methods, in terms of accuracy.

---

### Official Review · Reviewer_7fYz · 2021-11-02

**Correctness:** 4
**Technical Novelty And Significance:** 2
**Empirical Novelty And Significance:** 3
**Recommendation:** 8
**Confidence:** 3

**Main Review:**

PAPER SUMMARY: This paper presents a new compression algorithm for distributed machine learning, the algorithm is similar to QSGD, but uses a global scaling factor that is known to all parties. This adds additional benefits that the compressed results on each worker are addable.

NOVELTY & SIGNIFICANCE: Having compression results directly addable is a very useful property to support broader range of applications and improve the system performance.
The algorithm itself seems to be a simple trick based on QSGD (and indeed), but this simple trick seems to be quite effective. There are existing work on this topic (such as heuristic-SGD that was referenced in this paper). This paper adds theoretical analysis and better scaling factor choice.

TECHNICAL SOUNDNESS: The proof should be easy just following existing QSGD theoretical analysis with minor modifications. The only concern would be if the magnitude on different party varies a lot, there will be much larger compression error for the parties with smaller magnitude. But this shouldn't affect the (asymptotic) convergence rate.

OTHERS Experiments look fine.


**Summary Of The Paper:**

QSGD with compression results addable

**Summary Of The Review:**

This paper proposes a new compression algorithm for distributed deep learning where the compression results are addable). There are existing work on this topic, and the main contributions of this paper are providing theoretical analysis and giving better scaling factor choice.

---

> ### Author Response · Authors · 2021-11-23
> **Thank you for your positive feedback!**
>
> We thank the reviewer for the positive evaluation of our work!

---

### Official Review · Reviewer_q8za · 2021-11-03

**Correctness:** 3
**Technical Novelty And Significance:** 3
**Empirical Novelty And Significance:** 3
**Recommendation:** 8
**Confidence:** 2

**Main Review:**

The paper is clearly written, and the experiments show that the convergence issues with the nearest competitor (HeuristicSGD) are not just theoretical.  The main contribution really does seem to be the theoretical analysis of the convergence behavior.  The timing results show a 10-15% improvement over uncompressed distributed SGD (the all-reduce version); not a bad improvement, but not enormous, either.  The point remains that the proposed method requires only all-reduce communication (unlike many competitors) and does not require error feedback (unlike PowerSGD), and so does have a number of advantages over recently-published competitors.

I would have appreciated a little discussion of any shortcomings of the method, or next steps for future work.

Also, though I appreciated the experimental results, having a little more about the IntDIANA results in the body text (as opposed to the appendix) would have been more consistent with the claim that this is a main contribution of the paper.

Minor issues:

- "By combing (5) and (3)" -> "By combining (5) and (3)"

- "image classifcation" -> "image classification"

- I do not understand the red-blue color coding in some of the equations.

- In the experiments, are the data types float32 (for computation + SGD communication) and int8 for the intSGD communication?  This seems likely based on the communication timing, but it's hard to be sure.

**Summary Of The Paper:**

The paper introduces a randomized compress-to-integer operator with a shared scaling factor for use in data communication in distributed SGD.  The resulting algorithm is provably convergent, matches the behavior of SGD up to constant factors, and works well with the all-reduce primitive.  The authors claim three main contributions: the IntSGD algorithm itself, the analysis of convergence rates, and the IntDIANA variant for heterogeneous data distributions (though this is only discussed in the appendices).

**Summary Of The Review:**

The proposed algorithm represents an improvement in theory over the nearest competitor (the Heuristic IntSGD algorithm), requires only all-reduce style communication, and does not require error feedback.  Though the performance improvements in experiments were not enormous, there certainly were improvements.  I'd like to see the paper in ICLR.

---

> ### Author Response · Authors · 2021-11-23
> **We have revised the paper**
>
> We thank the reviewer for taking the time to read our paper and greatly appreciate their valuable feedback.
> > I would have appreciated a little discussion of any shortcomings of the method, or next steps for future work
>
> We thank the reviewer for this suggestion. One shortcoming of IntSGD is that the compression ratio is only 4x. In the future, we hope to find an improved compression technique that leads to a higher compression ratio while keeping the merits of the current algorithm (supports all-reduce, supports switch, EF-free, and fast compression). We have added a comment on this in the "Conclusion" section of our paper.
>
> > Having a little more about the IntDIANA results in the body text (as opposed to the appendix) would have been more consistent with the claim that this is a main contribution of the paper.
>
> We do agree that this discussion is missing in the main text, but we had to move it to the appendix due to the tight page limit.
>
> > Typos: "By combing (5) and (3)" -> "By combining (5) and (3)", "image classifcation" -> "image classification".
>
> Thank you for pointing out these typos. As suggested by another reviewer, we used a spell checker to find other typos in the paper and fixed 14 more small issues in the text.
>
> > I do not understand the red-blue color coding in some of the equations.
>
> The goal of the coloring schemes is to highlight the extra terms coming from our integer compression, which we give in red color, in contrast to the blue error terms, which come from SGD itself. We would prefer to keep the colors to help read the proofs (see, for instance, Lemma 2), but we can remove them if the reviewer suggests this.
>
> > In the experiments, are the data types float32 (for computation + SGD communication) and int8 for the intSGD communication?
>
> Yes, the reviewer is right, we used exactly these data types.

---

> > ### Comment · Reviewer_q8za · 2021-11-29
> > **After rebuttal**
> >
> > I thank the authors for their clarification and response to the other reviewers.  While I am not a champion for this paper, I still feel happier after this discussion, and have revised my score upward accordingly.

---

### Author Response · Authors · 2021-11-23
**To all reviewers**

Dear reviewers,

Thank you for the comments! We have updated our manuscript and provided detailed responses to address your concerns.

Authors

---

### Decision · Program_Chairs · 2022-01-20

**Decision:**

Accept (Spotlight)

**Comment:**

This paper proposes a theoretically sound and practically effective method to compress quantized gradients and reduce communication in distributed optimization. The method is interesting and worth publication.